# End-to-End Training Induces Information Bottleneck through Layer-Role Differentiation: A Comparative Analysis with Layer-wise Training

**Keitaro Sakamoto**                                    *sakakei-1999@g.ecc.u-tokyo.ac.jp*
*The University of Tokyo*

**Issei Sato**                                          *sato@g.ecc.u-tokyo.ac.jp*
*The University of Tokyo*

**Reviewed on OpenReview:** *https: // openreview. net/ forum? id= O3wmRh2SfT*

## Abstract

End-to-end (E2E) training, optimizing the entire model through error backpropagation, fundamentally supports the advancements of deep learning. Despite its high performance, E2E training faces the problems of memory consumption, parallel computing, and discrepancy with the functionalities of the actual brain. Various alternative methods have been proposed to overcome these difficulties; however, no one can yet match the performance of E2E training, thereby falling short in practicality. Furthermore, there is no deep understanding regarding differences in the trained model properties beyond the performance gap.

In this paper, we reconsider why E2E training demonstrates a superior performance through a comparison with layer-wise training, which shares fundamental learning principles and architectures with E2E training, with the granularity of loss evaluation being the only difference. On the basis of the observation that E2E training has an advantage in propagating input information, we analyze the information plane dynamics of intermediate representations based on the Hilbert-Schmidt independence criterion (HSIC). The results of our normalized HSIC value analysis reveal the E2E training ability to exhibit different information dynamics across layers, in addition to efficient information propagation. Furthermore, we show that this layer-role differentiation leads to the final representation following the information bottleneck principle. Our work not only provides the advantages of E2E training in terms of information propagation and the information bottleneck but also suggests the need to consider the cooperative interactions between layers, not just the final layer when analyzing the information bottleneck of deep learning.

## 1 Introduction

Backpropagation, which has been fundamentally supporting the recent advancements in machine learning, faces challenges in computational problems related to memory usage and parallel computing. Moreover, from a biological plausibility perspective, discrepancies with actual brain functions have been pointed out (Crick, 1989; Baldi et al., 2017). To mitigate these concerns, various learning algorithms and architectures have been proposed; however, there is still a performance gap between these alternatives and E2E training, which prevents practical use. A deeper understanding of the relationships between the E2E training method and non-E2E methods is significant, as it helps us set a direction for future research on learning methods without backpropagation. This will also lead to a reevaluation of the advantages of E2E training. Although the recent success of large language models (LLMs) such as GPT-4 is remarkable, the human brain has connections on the order of 100 trillion, whereas GPT-4 has roughly 1 trillion parameters (Stiles & Jernigan, 2010; OpenAI,

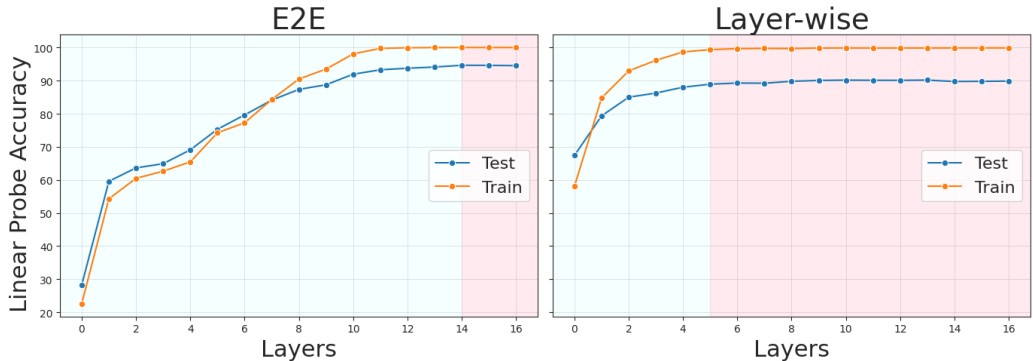

Figure 1: Linear probing accuracies of ResNet50 trained on CIFAR10 dataset. Linear separability for test data and training data are presented. **Left:** E2E training with cross-entropy loss. **Right:** Layer-wise training with cross-entropy loss. The model was trained with an auxiliary linear classifier for each block.

2023). This parameter efficiency prompts us to reconsider the biological plausibility in machine learning technologies, turning attention to the efficiency of current E2E training via backpropagation (Hinton, 2023).

To explore the properties of E2E training, we adopt layer-wise training for comparison, which shares the fundamental learning principles and architecture but differs in setting errors locally. As a first step, we confirm the performance gap between E2E and various layer-wise training methods and demonstrate that the performance of layer-wise training saturates with increasing depth, as illustrated in figure 1. This correlates with the finding of Wang et al. (2020), who showed that the greedy optimization in layer-wise training leads to the loss of input information required for classification at the early layers. This observation, coupled with the fact that E2E training can cooperatively learn each layer through backpropagation, motivates the information-theoretic analysis of the trained model. Specifically, we analyze the mutual dependence among the model input, class labels, and intermediate representations, following the information bottleneck work (Tishby et al., 2000; Tishby & Zaslavsky, 2015). We investigate with normalized HSIC (Fukumizu et al., 2007), which is used to analyze the neural network in several existing studies, to circumvent the difficulty in estimating mutual information (Ma et al., 2020; Wang & Isola, 2020; Wang et al., 2023b). Our analysis centered around HSIC dynamics reveals the two following findings:

1. The E2E-trained model obtains the middle representations with a higher correlation with labels, despite having lower class separability. We present a toy example, independently of the HSIC analysis, where layer-wise training causes the degeneration in mutual information. These analyses imply that E2E training is advantageous in information propagation.

2. E2E training compresses middle representations maintaining a high HSIC value in the final layer for relatively deep models. In contrast, layer-wise training shows uniform compression in every layer or uniform increase with the method to keep input information as in Wang et al. (2020). The layer-wise training shows the same HSIC behavior at each layer, whereas E2E training can assign different roles of information compression among layers. We refer to this property as *layer-role differentiation*. The term "differentiation" refers to the process through which different layers develop distinct roles or functions and is different from the mathematical operation of differentiation in calculus.

Additionally, our HSIC analysis shows the information bottleneck behavior in the final layer only for E2E training. It shows the benefit of E2E training in terms of information bottleneck and suggests a connection between the middle layer compression in E2E training brought by layer-role differentiation and the acquisition of information bottleneck representation at the final layer. Finally, as another advantage of intermediate layer compression, we derive the connection with Frosst et al. (2019), claiming that the class entanglement in the middle layers leads to a better representation in the final layer. Since both E2E and layer-wise training

shares the same backbone architecture, note that our analysis is not about the benefits of deepening the layers but is about the advantages of training in the E2E manner. Our code is available on GitHub [1].

## 2 Difference between E2E and Layer-wise Training

We first provide a clear definition and problem setting for E2E training and layer-wise training. Subsequently, layer-wise training methods, which have some variations depending on the type of local loss and architectures, are organized and compared. We show that while there are differences in performance among these methods, the persisting performance gap remains compared to E2E training.

### 2.1 Problem Setting

First, let us clarify the definitions of End-to-end (E2E) training and layer-wise training. E2E training refers to a learning method where the objective function for the entire model is set, and the weights gradient updates are propagated throughout the entire model using backpropagation. In contrast, layer-wise training is a learning approach to set an objective function for each layer. The influence of the local objective function is confined to the specific layer alone, and gradient information does not propagate to the preceding layers. In layer-wise training, each layer is trained simultaneously, and the model's prediction is made by using only the last output. We consider the local module size to be one layer; however, in cases where straightforward layer-wise modularization is difficult, such as a skip-connection in the ResNet blocks, we handle the natural minimum module that includes one skip-connection (see section B.1 for training details).

Next, we organize the problem settings and define the notations in the paper. We consider a multi-class classification problem with a dataset $\{x_i, y_i\}_{i=1}^m$ consisting of inputs $x_i \in \mathcal{X}$ and the class label $y_i \in \mathcal{Y}$. Specifically, we consider the domain $\mathcal{X} = \mathbb{R}^d, \mathcal{Y} = \{1, \ldots, c\}$, where $c$ is the number of classes, and use $\boldsymbol{x}$ to express the vector input explicitly. The dataset is sampled from the underlying distribution $p_{XY}$, and we denote the input random variables and label random variables as $X$ and $Y$.

We have $L$ layers $f_1, \ldots, f_L$ and stack them to form a single neural network $f$. The dimensions of the $l$-th layer's representation are denoted as $d_l$, i.e., $l$-th layer is $f_l : \mathbb{R}^{d_{l-1}} \to \mathbb{R}^{d_l}$. Note that the intermediate representations are allowed to be more than one-dimensional, but they are vectorized on this notation. The $l$-th layer's representation for input $\boldsymbol{x}$ is denoted as $\boldsymbol{z}_l$, meaning $\boldsymbol{z}_l = f_l \circ \cdots \circ f_1(\boldsymbol{x})$. In cases where the input is indexed, the $l$-th representation for $\boldsymbol{x}_i$ is written as $\boldsymbol{z}_{l,i}$. We simply use $\boldsymbol{z}$ to denote the middle representation without specifying the layer explicitly. To clear the optimized weights, let us denote the weights of each layer $f_l$ by $\boldsymbol{\theta}_l$, which are denoted as $\boldsymbol{\theta}$ collectively. The objective loss for E2E learning is

$$\mathcal{L}_{E2E}(\boldsymbol{\theta}) = \frac{1}{m} \sum_{i=1}^m \ell(f(\boldsymbol{x}_i), y_i), \tag{1}$$

where $\ell$ is the loss function that takes the model outputs and the true label. The cross-entropy loss is widely used for multi-class classification. In contrast, the loss to optimize for layer-wise training is

$$\mathcal{L}_{LW}(\boldsymbol{\theta}) = \sum_{l=1}^L \mathcal{L}_l(\boldsymbol{\theta}_l) = \sum_{l=1}^L \left( \frac{1}{m} \sum_{i=1}^m \ell_l \left( f_l(\text{StopGrad}(\boldsymbol{z}_{l-1,i})), y_i \right) \right), \tag{2}$$

where StopGrad is the operator to stop gradient propagation, such as "Tensor.detach()" in the PyTorch implementation, which enables layer-wise training. Here, the loss of the $l$-th layer is denoted as $\ell_l$; however, note that we use the same loss $\ell$ among the layers in this study. When calculating this local loss, various layer-wise training methods adopt auxiliary networks for each layer. While the algorithm that directly optimizes the representation space does not need these networks, in the case of classification loss such as cross-entropy, the loss is evaluated after projecting onto $c$ dimensional space. We denote the auxiliary network used in the $l$-th layer by $g_l : \mathbb{R}^{d_l} \to \mathbb{R}^{d_l'}$, which can be a linear layer, multi-layer perceptron (MLP), or identity mapping in the case of no auxiliary networks. Additionally, the output of the auxiliary networks is denoted as $\boldsymbol{h}_1 \cdots, \boldsymbol{h}_L$. The problem settings of E2E training and layer-wise training are summarized in figure 2.

---
[1] https://github.com/keitaroskmt/E2E-info

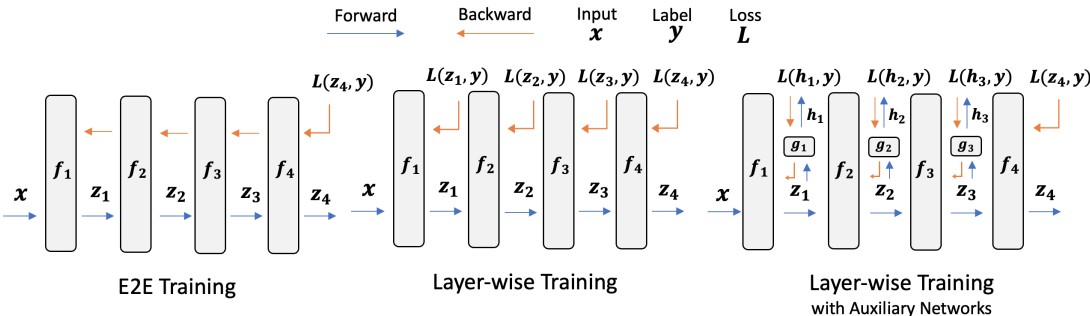

Figure 2: Comparison between E2E training and layer-wise training. The models have four layers, and each layer is denoted as $f_1, f_2, f_3, f_4$. **Left:** E2E training evaluates the loss only after the last representation $z_4$. The gradient updates are back-propagated to the preceding layers. **Middle:** Layer-wise training calculates the loss for each layer's outputs and updates weights. **Right:** Layer-wise training with the auxiliary networks.

## 2.2  Performance Gap

In this section, we empirically explore the performance gap between E2E training and several methods of layer-wise training. Layer-wise training has several variations based on the type of locally defined loss function and the auxiliary networks' types and sizes. We begin by organizing these methods and elucidating their performance differences compared to E2E learning methods.

In the layer-wise training methods, locally used loss falls into two primary types based on the method of providing the label information $y$. The first optimizes the classification loss at each layer by forwarding input **x** only. The second one forwards both the input **x** and its corresponding input, called the prototype, representing the class label information of **x**, and optimizes losses based on spatial relationships in each layer's representation. For the former case, we use cross-entropy loss (CE), while for the latter, supervised contrastive loss (SupCon) (Khosla et al., 2020) and similarity matching loss (Sim) (Nøkland & Eidnes, 2019) are adopted for the case where random images are used as prototypes, and signal propagation (SP) (Kohan et al., 2023) is used for the case where the class itself is projected onto the input space and used as a prototype. Please refer to section B.2 for the details of these methods.

The comparison of these methods' accuracies for CIFAR10 is presented in Table 1, and for the CIFAR100 dataset, they are detailed in Table 4 in the appendix. We used three networks with different sizes: VGG11, ResNet18, and ResNet50. Here, please note that the layer-wise training without auxiliary networks corresponds to the row of the 0-layer. The experimental details are also deferred to section F.1. These results indicate two significant observations. First, the performance of layer-wise training was worse when directly optimizing middle layers without auxiliary networks, and was greatly enhanced with the existence or the size of them. However, there remains an accuracy gap compared to E2E training across all loss functions. When the network had already achieved a relatively high accuracy with an auxiliary network with one layer, adding a second layer did not yield significant improvements, and the model was not competitive with the one trained in the E2E way. Second, in E2E training, there was a substantial improvement in accuracy by increasing the depth of networks in the order of VGG11, ResNet18, and ResNet50. Conversely, the amount of accuracy improvement was limited in the case of layer-wise training. As in the size of auxiliary networks, increasing the number of layers in the backbone model did not lead to a significant accuracy improvement.

## 2.3  Linear Separability

Then, what properties of the trained models contribute to such performance differences? A key distinction between E2E and layer-wise training lies in how the behavior of the intermediate layers is specified. In E2E training, this behavior is indirectly specified, whereas, in layer-wise training, it is directly controlled through the optimization of local losses. For instance, when optimizing classification errors at each layer using linear auxiliary classifiers, the training aims to enhance the linear separability of intermediate representations. Contrasting this with the E2E training provides insight into the property differences. Figure 1 compares

Table 1: Test accuracies of E2E training and layer-wise training methods trained on CIFAR10 dataset. The "Auxiliary" column shows the size of auxiliary networks $g$ for layer-wise training. For clarity, the highest accuracy among the different sizes of auxiliary networks is indicated in bold. For CE loss, the middle representations are converted to space of class numbers' dimensions; therefore, it does not have 0-layer case.

| Model | Auxiliary | E2E | | Layer-wise | | | | |
|---|---|---|---|---|---|---|---|---|
| | | CE | SupCon | CE | SupCon | Sim | SP (Hard) | SP (Soft) |
| VGG11 | 0-layer | | | — | 60.6 | 72.4 | 82.4 | 81.5 |
| | 1-layer | 91.5 | 90.4 | **89.3** | 83.7 | **89.2** | — | — |
| | 2-layer | | | 88.4 | **89.1** | 89.0 | — | — |
| ResNet18 | 0-layer | | | — | 74.0 | 87.8 | 58.0 | 45.9 |
| | 1-layer | 93.9 | 95.7 | 89.3 | 88.1 | 89.4 | — | — |
| | 2-layer | | | **89.9** | **92.5** | **91.6** | — | — |
| ResNet50 | 0-layer | | | — | 73.2 | 84.7 | 51.7 | 19.5 |
| | 1-layer | 94.7 | 96.3 | 89.9 | 88.7 | 91.2 | — | — |
| | 2-layer | | | **90.4** | **92.0** | **91.3** | — | — |

the linear separability measured through linear probing (Alain & Bengio, 2016) at each layer. In the E2E case, the linear separability gradually improves, whereas, in layer-wise training, it abruptly improves in the early layers and then saturates at the same level as depicted in the red region. Both approaches ultimately achieve perfect separation on the training data, but E2E demonstrates better linear separability on the test data. Interestingly, in layer-wise training, stacking layers does not improve the separability of the final layer. This is consistent with previous experiments that show that the increase in the model size did not lead to as much accuracy improvement as observed in the E2E training. This holds true for similarity-based local loss functions, and the results for the SupCon case are presented in section F.3.

We further investigated why the layer-wise training does not benefit from stacking layers. The layer-wise trained model was retrained in the E2E manner after the layer where the linear separability starts to saturate, i.e., the border between blue and red regions in figure 1. Our aim here was to observe test accuracy improvement, which would potentially indicate whether valuable input information remains in these intermediate layers that layer-wise training does not fully utilize. As indicated in Table 2, the outcomes of this experiment under the same setting as figure 1 demonstrated only marginal accuracy improvement

Table 2: Retrain results of ResNet50, CIFAR10. The same trained models as figure 1 are used.

| | E2E | LW | Retrain |
|---|---|---|---|
| Test acc | 94.8 | 89.9 | 90.8 |

through E2E retraining. This suggests that useful input information for classifying is lost even in the early layers due to localized optimization, reconfirming the information collapse hypothesis suggested by (Wang et al., 2020). In the following sections, we explore the information propagation within the trained model, elucidating the role of E2E training.

## 3 Information Bottleneck for Analyzing Deep Neural Network

The comparison between E2E and layer-wise training in the previous section motivates us to evaluate the information flow between layers. We focus on the concept of *information bottleneck* to deal with the change of mutual information within the trained model. In this section, let us introduce the related concepts and normalized HSIC as an alternative to mutual information.

### 3.1 Information Bottleneck

Information Bottleneck (IB) (Tishby et al., 2000) refers to the compression of the input $X$ to obtain the representation $Z$ with the help of the random variable $Y$ behind $X$, in this case, the label information. It uses mutual information among these random variables to compress $X$ while trying to retrain as much

information about $Y$ contained in $X$ as possible during the compression process. In recent years, IB has been used to analyze the dynamics of deep learning (Tishby & Zaslavsky, 2015; Saxe et al., 2018). IB aims to extract the task-relevant information from $X$ as much as possible, by minimizing

$$I(X;Z) - \beta I(Y;Z), \tag{3}$$

where $\beta > 0$ is the trade-off parameter and $I$ denotes the mutual information (see section D for the definition). The task-irrelevant information in $X$ is squeezed by minimizing this Lagrangian objective, and the random variable $Z$ can be regarded as a compression of $X$ to solve the task. In this framework, the dynamics of the final representation in deep learning are explained by learning the compressed representation $Z$ that minimizes the IB objective. From the IB perspective, multilayered neural networks can be viewed as a sequence of data processing steps. The input $X$ is generated from the label $Y$ behind $X$ and transformed into the representations $Z_1, Z_2, \ldots$ as it passes through each layer. This process can be represented by the Markov chain $Y \to X \to Z_1 \to \cdots \to Z_L$. From the data processing inequality,

$$I(Y;X) \geq I(Y;Z_1) \geq \cdots \geq I(Y;Z_L). \tag{4}$$

Intuitively, the compressed representation $Z$ cannot have more information about the label $Y$ than the original input $X$. The network aims to transform $X$ without losing much information from the given upper bound $I(Y;X)$, extracting useful features from $X$ and enhancing the class separability in the final layer. Moreover, considering the reverse data processing inequality, it holds that

$$I(Y;Z) \leq I(X;Z) \tag{5}$$

for any middle representations $Z$. The IB hypothesis states that in the final layer, there is an initial phase where $I(Y;Z)$ increases along with $I(X;Z)$, followed by a phase that reduces $I(X;Z)$ to compress task-irrelevant information. This implies that while increasing the lower bound in equation 5 raises $I(X;Z)$, the network training subsequently reduces $I(X;Z)$ to minimize the gap with the lower bound.

Analyzing mutual information within the deep neural network is an intriguing research area. However, there are two major challenges: 1) the potential for $I(X;Z)$ to become infinite, and 2) the difficulty of estimation due to the high dimensionality. For more details, please refer to section D.2 in the appendix.

## 3.2  HSIC as an alternative to Mutual Information

Considering the difficulty in estimating mutual information, we use the Hilbert-Schmidt independence criterion (HSIC) (Gretton et al., 2005) because it also allows the capture of non-linear relationships of random variables in a non-parametric way. The use of HSIC is supported by the fact that it has a relationship with a variant of mutual information and the previous studies (see related work, section 6).

The HSIC is a statistical measure of the dependence between two random variables. Suppose we have two random variables $X, Y$ on probability spaces $(\mathcal{X}, P_X), (\mathcal{Y}, P_Y)$, and their corresponding reproducing kernel Hilbert spaces (RKHSs) are $\mathcal{H}_{\kappa_1}, \mathcal{H}_{\kappa_2}$ with measurable positive definite kernels $\kappa_1 : \mathcal{X} \times \mathcal{X} \to \mathbb{R}, \kappa_2 : \mathcal{Y} \times \mathcal{Y} \to \mathbb{R}$. The mean $\mu_X$ is an element of $\mathcal{H}_{\kappa_1}$ such that $\langle \mu_X, f \rangle = \mathbb{E}_X[f(X)]$ for all $f \in \mathcal{H}_{\kappa_1}$. Let $\mu_Y, \mu_{XY}$ denote the mean element on $\mathcal{H}_{\kappa_2}, \mathcal{H}_{\kappa_1} \otimes \mathcal{H}_{\kappa_2}$ respectively. The cross-covariance operator $\mathcal{C}_{XY} : \mathcal{H}_{\kappa_2} \to \mathcal{H}_{\kappa_1}$ is defined by $\mathcal{C}_{\mathcal{X}\mathcal{Y}} := \mu_{XY} - \mu_X \mu_Y$. Note that $\mu_{XY} - \mu_X \mu_Y$ is an element of $\mathcal{H}_{\kappa_1} \otimes \mathcal{H}_{\kappa_2}$, and it can be regarded as a linear operator from $\mathcal{H}_{\kappa_2}$ to $\mathcal{H}_{\kappa_1}$, defined as $\langle f, \mathcal{C}_{XY} g \rangle_{\mathcal{H}_{\kappa_1}} = \langle \mu_{XY} - \mu_X \mu_Y, fg \rangle_{\mathcal{H}_{\kappa_1} \otimes \mathcal{H}_{\kappa_2}}$, for any $f \in \mathcal{H}_{\kappa_1}, g \in \mathcal{H}_{\kappa_2}$.

The Hilbert Schmidt norm of the operator $\mathcal{C} : \mathcal{H}_{\kappa_2} \to \mathcal{H}_{\kappa_1}$ is defined as $\|\mathcal{C}\|_{HS} = \sqrt{\sum_{i,j} \langle \phi_i, \mathcal{C}\psi_j \rangle_{\mathcal{H}_{\kappa_1}}}$, where $\{\phi_i\}$ and $\{\psi_i\}$ are the orthogonal basis of $\mathcal{H}_{\kappa_1}$ and $\mathcal{H}_{\kappa_2}$, respectively. HSIC is defined by the Hilbert Schmidt norm of the cross-covariance operator,

$$\mathrm{HSIC}(X, Y) := \|\mathcal{C}_{XY}\|_{HS}^2. \tag{6}$$

Since this equals to $\|\mu_{XY} - \mu_X \mu_Y\|_{\mathcal{H}_{\kappa_1} \otimes \mathcal{H}_{\kappa_2}}^2$, the value of the HSIC can be computed specifically as follows:

$$\begin{aligned}
\mathrm{HSIC}(X, Y) = {} & \mathbb{E}_{XYX'Y'}[\kappa_1(X, X')\kappa_2(Y, Y')] - 2\mathbb{E}_{XY}[\mathbb{E}_{X'}[\kappa_1(X, X')|X]\mathbb{E}_{Y'}[\kappa_2(Y, Y')|Y]] \\
& + \mathbb{E}_{XX'}[\kappa_1(X, X')]\mathbb{E}_{YY'}[\kappa_2(Y, Y')],
\end{aligned} \tag{7}$$

where $(X', Y')$ is independent from $(X, Y)$ and follows the identical distribution.

Given finite samples $\{(x_1, y_1), \ldots, (x_m, y_m)\}$ following the joint distribution $P_{XY}$, one can give an empirical estimator of HSIC as $\text{Tr}(\boldsymbol{K}_1 \boldsymbol{H} \boldsymbol{K}_2 \boldsymbol{H})/(m-1)^2$, where $\boldsymbol{K}_1, \boldsymbol{K}_2, \boldsymbol{H} \in \mathbb{R}^{m \times m}, K_{1ij} = \kappa_1(x_i, x_j), K_{2ij} = \kappa_2(y_i, y_j), H_{ij} = \delta_{ij} - 1/m$. In this work, we use the normalized HSIC (nHSIC), which is a normalized version of the HSIC. It equals the chi-squared divergence between $P_{XY}$ and $P_X P_Y$, which is called the chi-squared mutual information (see section D.3). The nHSIC is defined by the Hilbert Schmidt norm of the normalized cross-covariance operator, $\text{nHSIC}(X, Y) \coloneqq \|\mathcal{C}_{XX}^{-1/2} \mathcal{C}_{XY} \mathcal{C}_{YY}^{-1/2}\|_{HS}^2$. The empirical estimator for nHSIC has consistency (Fukumizu et al., 2007); however, the unbiased estimate is not calculated analytically. Regarding the choice of kernels and the estimate for nHSIC, we followed existing information bottleneck analyses that use nHSIC (Ma et al., 2020; Wang et al., 2021; 2023b). RBF kernels were used for the input variables $X$ and the intermediate variables $Z$ to capture non-linear relationships. As for the class categorical variable $Y$, a linear kernel was applied because it is a discrete random variable (see section F.1 for details).

# 4  HSIC Dynamics of E2E and Layer-wise Training

As observed in section 2, the input information required to solve the task might be lost in the early layers in the layer-wise training case. If $I(X; Z)$ is not significantly increased in the initial layers, then according to equation 5, $I(Y; Z)$ does not sufficiently improve either. Consequently, $I(Y; Z)$ presumably fails to become large in the subsequent layers and eventually in the last layer. This is not desirable behavior in terms of the information flow within the whole model. Additionally, it is uncertain whether an independent training of each layer induces a compression phase that reduces $I(X; Z)$ while preserving $I(Y; Z)$ in the final layer, which is suggested by the IB hypothesis.

Under these motivations, we aim to analyze the difference in the dynamics of $I(X; Z)$ and $I(Y; Z)$, i.e., the information plane, between E2E training and layer-wise training. We first experiment on the smaller settings where the difference starts to appear and explain the mechanism behind the observation. We then move to the same settings as section 2 in the subsequent sections. As mentioned in the previous section, while our discussion is centered around the mutual information, we experiment with $\text{nHSIC}(X, Z)$ and $\text{nHSIC}(Y, Z)$ to show the information plane.

## 4.1  HSIC Plane Dynamics for LeNet5 Setting

Firstly, we compared the dynamics of $\text{nHSIC}(X, Z)$ and $\text{nHSIC}(Y, Z)$ between E2E training and layer-wise training using a simple model. Figure 3 illustrates the results for the LeNet5 model (LeCun et al., 1998) for 100 epochs, and the training progression is depicted using a color bar and arrows. The LeNet5 is composed of five layers, with the initial two convolutional layers and the latter three linear layers, and for the layer-wise training, linear auxiliary networks are adopted for each layer.

Figure 3a shows the results for the MNIST dataset, where it is clear that there is not so much difference in the HSIC plane dynamics between E2E training and layer-wise training. Furthermore, the actual test accuracy of the trained models showed little difference as in the caption of figure 3a. This outcome suggests that for the MNIST problem setting, where the local optimization already works reasonably well, even the greedy optimization in each layer could learn representation $Z$ that has a high correlation with $X$ and $Y$. Next, Figure 3b shows the results for the CIFAR10 dataset, which is the larger setting. In this setup, there is a noticeable gap in the test accuracy between E2E and layer-wise training. Moreover, an intriguing difference was observed in the HSIC dynamics. In E2E training, both $\text{nHSIC}(X, Z)$ and $\text{nHSIC}(Y, Z)$ show a consistent increase in all layers as seen in the MNIST setting. However, in the first layer of layer-wise training, the HSIC values increase at first and subsequently turn to decrease. While this tendency was not observed in the later layers, the layer-wise training did not obtain representations with HSIC values as high as those in E2E learning.

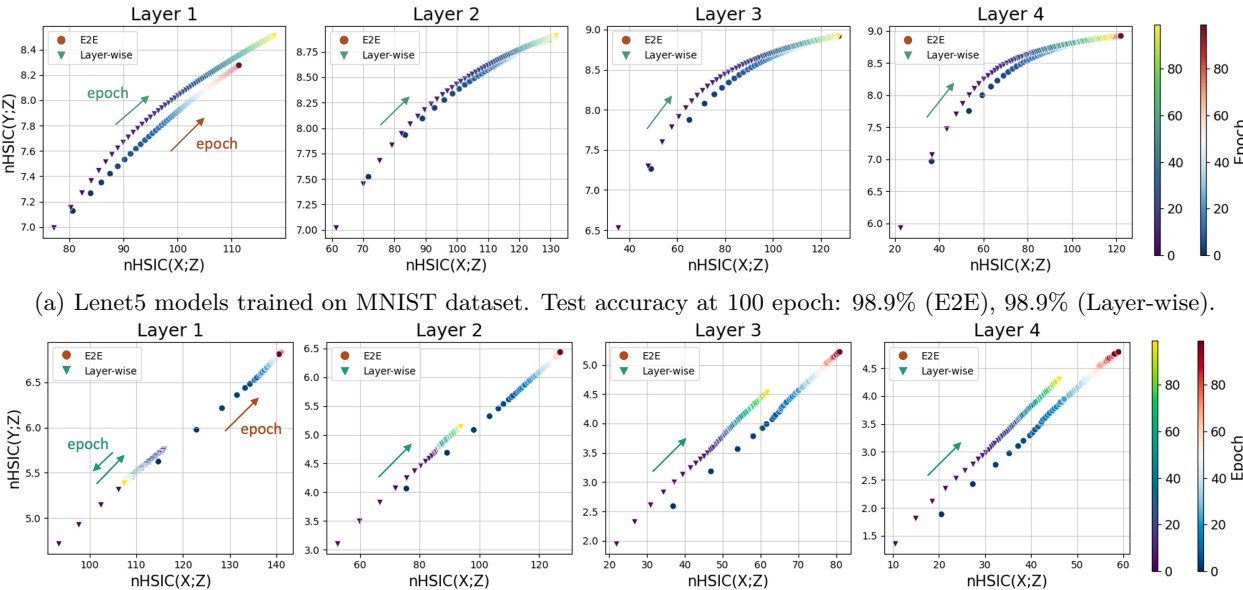

(a) Lenet5 models trained on MNIST dataset. Test accuracy at 100 epoch: 98.9% (E2E), 98.9% (Layer-wise).

(b) Lenet5 models trained on CIFAR10 dataset. Test accuracy at 100 epoch: 62.5% (E2E), 59.2% (Layer-wise).

Figure 3: HSIC plane dynamics of LeNet5 model. The color gradation shows the progress of training, i.e., the number of epochs. Inverted triangles with a blue-yellow-based colormap denote layer-wise training, whereas circles with a red-based colormap show E2E training.

## 4.2 Example of Greedy Optimization Failure

If the HSIC value achieved in the final layer accounts for the performance discrepancy between E2E and layer-wise training, understanding how this difference arises could shed light on the shortcomings of layer-wise training and further highlight the strength of E2E training. We observed that the HSIC value for layer-wise training is lower even in the initial layer and turns into an intriguing decrease. Considering that each layer is provided the class information explicitly in the layer-wise training, such an HSIC behavior is not necessarily intuitive. In this section, we demonstrate how such a phenomenon might arise using a toy example. When we focus on the initial representation $Z_1$, there is no difference in the incoming inputs between E2E and layer-wise training. The distinction lies in the distance to the loss evaluation: in layer-wise training, it occurs after one linear layer, while in E2E training, it happens after the subsequent backbone model.

We first see how the optimization of cross-entropy loss is related to mutual information with label $Y$. From the non-negativity of KL divergence and the Markov chain $Y \rightarrow X \rightarrow Z$, we obtain the lower bound of the mutual information as follows (see section E.1 for detail):

$$I(Y;Z) \geq \mathbb{E}_{p(y,\boldsymbol{x})}\left[\mathbb{E}_{p(\boldsymbol{z}|\boldsymbol{x})}\left[\log q(y|\boldsymbol{z})\right]\right] + H(Y), \tag{8}$$

where $q(\mathrm{y}|\mathbf{z})$ is an arbitrary variational distribution and $H(Y)$ is the entropy of the label distribution. The bound is tight when $q(\mathrm{y}|\mathbf{z}) = p(\mathrm{y}|\mathbf{z})$ holds. The $p(\boldsymbol{z}|\boldsymbol{x})$ is determined by the backbone model up to the layer, and in the layer-wise training, the variational distribution $q(y|\boldsymbol{z})$ is modeled by the auxiliary classifier. Optimizing the cross-entropy loss calculated from its output is equivalent to maximizing the lower bound of equation 8. The layer-wise training, which can only train the final layer of $p(\boldsymbol{z}|\boldsymbol{x})$ and a linear classifier $q$, could not significantly maximize the lower bound due to the insufficient capacity of the model to represent the relationship between $X$ and $Y$. Therefore, learning representations with a high $I(Y;Z)$ value becomes challenging. Furthermore, as will be discussed next, the local optimization in this scenario could lead to inappropriate learning of the backbone model for the later layers.

Let us consider the toy data model in figure 4 as an example of $p(\mathrm{y}|\mathbf{x})$ having non-linear relationships. Note that in the E2E training setting, which allows non-linear transforms, perfect class separation is possible based on the length of the radius $r$. We consider a binary classification setup where $\boldsymbol{x} \in \mathbb{R}^2$ follows a distribution

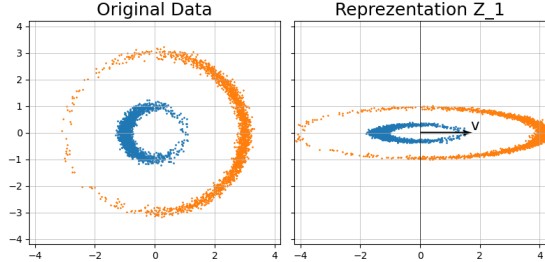
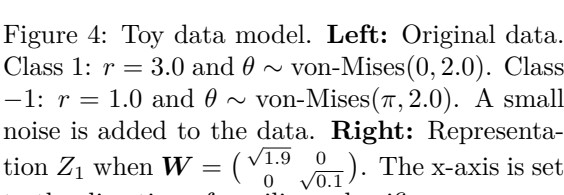
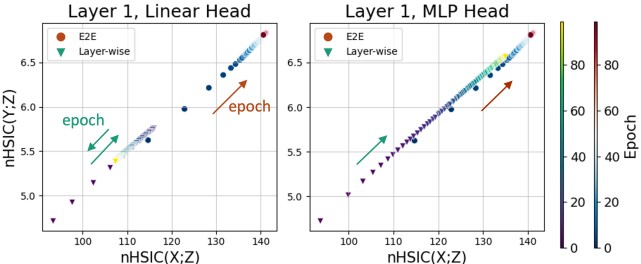

Figure 4: Toy data model. **Left:** Original data. Class 1: $r = 3.0$ and $\theta \sim$ von-Mises$(0, 2.0)$. Class $-1$: $r = 1.0$ and $\theta \sim$ von-Mises$(\pi, 2.0)$. A small noise is added to the data. **Right:** Representation $Z_1$ when $\boldsymbol{W} = \left( \begin{smallmatrix} \sqrt{1.9} & 0 \\ 0 & \sqrt{0.1} \end{smallmatrix} \right)$. The x-axis is set to the direction of auxiliary classifier $\boldsymbol{v}$.

Figure 5: HSIC plane dynamics of the initial layer of Lenet5 models trained on CIFAR10 dataset. The size of auxiliary networks is different; the left shows the results for the linear head (same as figure 3b), and the right shows the results for the two-layer MLP head.

illustrated in figure 4. The representation of the first layer $Z_1$ is determined as: $\boldsymbol{z}_1 = f_1(\boldsymbol{x}) = \boldsymbol{W}\boldsymbol{x}$, where $\boldsymbol{W} \in \mathbb{R}^{2 \times 2}$ is constrained by $\|\boldsymbol{W}\|_F \leq c$; $c$ is a constant. Additionally, the auxiliary classifier $g$ is linear as $g_1(\boldsymbol{z}_1) = \boldsymbol{v}^\top \boldsymbol{z}_1$, where $\|\boldsymbol{v}\|_2 = 1$, and the class is predicted based on the output sign. In the layer-wise training, the binary cross-entropy is optimized based on the output of $g_1$. Adding the constraint term on $\boldsymbol{W}$, the loss function for the first layer is as follows:

$$\mathcal{L}_{LW}(\boldsymbol{W}, \boldsymbol{v}) = \frac{1}{m} \sum_{i=1}^{m} \log\left(1 + \exp\left(-2y_i \boldsymbol{v}^\top \boldsymbol{W} \boldsymbol{x}_i\right)\right) + \lambda \|\boldsymbol{W}\|_F^2, \tag{9}$$

where $\lambda > 0$ is the hyperparameter. The gradients for $\boldsymbol{v}$ and $\boldsymbol{W}$ are

$$\frac{\partial \mathcal{L}_{LW}}{\partial \boldsymbol{v}} \propto \sum_{i=1}^{m} -y_i \boldsymbol{W} \boldsymbol{x}_i, \quad \frac{\partial \mathcal{L}_{LW}}{\partial \boldsymbol{W}} \propto \sum_{i=1}^{m} -y_i \boldsymbol{v} \boldsymbol{x}_i^\top + 2\lambda \boldsymbol{W}. \tag{10}$$

The data symmetry implies that $\sum_{i:y_i=1} \boldsymbol{x}_i$ points in the positive direction of the x-axis, while $\sum_{i:y_i=-1} \boldsymbol{x}_i$ points in the negative direction of the x-axis. Consequently, $\sum_i y_i \boldsymbol{x}_i$ aligns with the positive direction of the x-axis. As a result, based on the gradient update equation 10, the second column of $\boldsymbol{W}$, $W_{1,2}$ and $W_{2,2}$, converges to zero due to norm regularization. Figure 4 illustrates the process of rank collapse during training. In this figure, the basis is changed so that the auxiliary classifier $\boldsymbol{v}$ aligns with the positive direction of the x-axis. The first column of $\boldsymbol{W}$ points in the same direction as $\boldsymbol{v}$ in this figure, which follows from the gradient update equation 10 for small $\lambda$ intuitively, but the alignment is not necessarily required for this discussion. The important point here is the rank collapse of $\boldsymbol{W}$, leading to representations that more closely adhere to the x-axis. After the collapse, when observing points near the origin on the x-axis, we cannot accurately distinguish their corresponding labels or original data points. This situation results in a reduction in $I(Y; Z)$ and $I(X; Z)$ because the uncertainty in $Y$ and $X$ arises after observing the realization of $Z$.

On the basis of this analysis, we hypothesize the following two steps could occur in the increasing and decreasing phases of HSIC in figure 3b; First, in the early phase of training, the representation is trained to be correlated with the label. Next, as training progresses, a decrease in mutual information similar to that in the toy data example occurs. In actual computing, a decline happens when representation becomes indistinguishable in terms of the computational precision of the floating point even before a complete collapse of representation. Although this analysis provides insights only into a linear auxiliary head case, there exists a performance gap between E2E and layer-wise training even with two-layer MLP as seen in section 2. Figure 5 illustrates the HSIC dynamics of the first layer when using a two-layer MLP instead of a linear auxiliary classifier, under the same LeNet5 and CIFAR10 setup as in the previous section. This change effectively resolves the decrease in the HSIC, leading to a similar increasing trend as in E2E training. However, the fact that E2E training achieves better test accuracy in the setting of table 1 suggests that the two-layer MLP might be insufficient as a model for $q(y|\boldsymbol{z})$, potentially due to its small size or not being a convolutional network, leading to such a collapse of input information with greedy optimization.

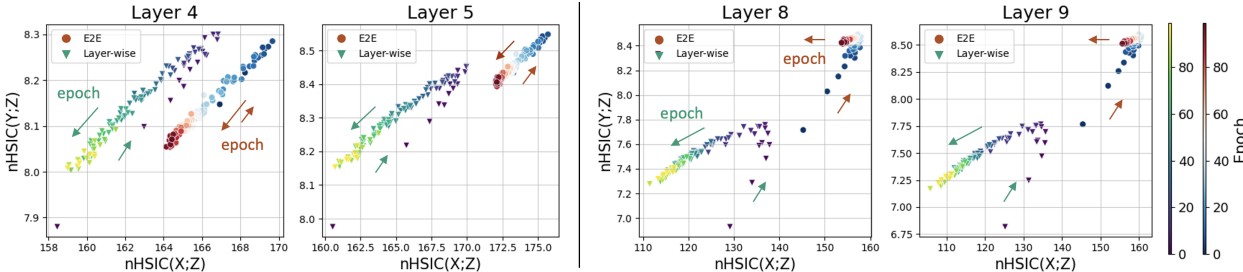

Figure 6: HSIC plane dynamics of ResNet18 model trained on CIFAR10. **Left:** Middle layers. **Right:** Output layers. The color gradation shows the progress of training. Inverted triangles with a blue-yellow-based colormap denote layer-wise training, whereas circles with a red-based colormap show E2E training.

## 4.3 HSIC Plane Dynamics for ResNet Setting

Thus far, we have investigated relatively small models. In this section, we present the HSIC dynamics of ResNet. Figure 6 shows the results for the CIFAR10 dataset. While ResNet18 consists of nine middle representations from the first to the penultimate layer, the middle layers and output layers are shown in the figure. Please refer to figure 15 in the appendix for all layers' results. In the case of LeNet5 settings, both E2E and the layer-wise training generally showed an increase in HSIC values during training, while in the ResNet setting, both approaches demonstrated a decrease of nHSIC$(X, Z)$ and nHSIC$(Y, Z)$ in the middle layers of the network. However, they do not have similar HSIC plane dynamics for every layer. In the output-side layers, E2E training showed a slight compression phase of nHSIC$(X, Z)$ while maintaining a high nHSIC$(Y, Z)$, aligning with the compression phase suggested by the IB theory. In contrast, in the layer-wise approach, both nHSIC$(X, Z)$ and nHSIC$(Y, Z)$ decreased as in the intermediate layers. Note that the value of nHSIC$(Y, Z)$ itself does not necessarily imply linear separability; therefore, the actual test accuracy did not deteriorate during this HSIC decreasing phase. On the basis of the findings, the advantages of E2E learning over layer-wise training can be attributed to the following two main factors. The results of other models are presented in section F.4, and similar behavior was observed. In particular, the IB behavior in the final layer was more distinct in ResNet50 and VGG11 with batch normalization.

**Efficient learning of $Z$ with high HSIC.** E2E training can acquire the representation with high HSIC at the very early phase of training. This suggests that learning with back-propagation is more efficient from the perspective of HSIC than directly providing class information to each layer, as in the layer-wise training.

**Layer-role differentiation.** While both E2E and layer-wise approaches exhibit the HSIC decrease phase in the middle layers, there is a difference in the mechanism behind this phenomenon. In the layer-wise training, due to repeating similar optimization at each layer, even the output layers showed the same HSIC behaviors. However, in the E2E training, the reduction in HSIC occurs in the middle layers while maintaining high nHSIC$(Y, Z)$ in the output layers. E2E can compress the middle layers while guaranteeing a certain level of $I(Y; Z_L)$ from equation 4. In contrast, layer-wise training lacks knowledge of the later layers during optimization, and each layer's optimization limits the upper bound of $I(Y; Z_L)$. These observations imply that the advantages of being E2E, i.e., enabling interaction among layers, is having different information-theoretical roles among layers and acquiring information bottleneck representation at the last layer.

## 4.4 Controlling HSIC in Layer-wise Training

The observations in the preceding sections have indicated that the layer-wise training can cause a reduction phase in HSIC through greedy optimization at each layer, potentially hindering the information propagation to subsequent layers. In this section, we explore the possibility that enhancing nHSIC$(X, Z)$ during local optimization enables layer-wise training to achieve a model performance comparable to E2E training. Wang et al. (2020) incorporated the reconstruction error, which appears in the variational lower bound of $I(X; Z)$, into the local error to mitigate the reduction of $I(X; Z)$ due to local optimization. Although we differ in

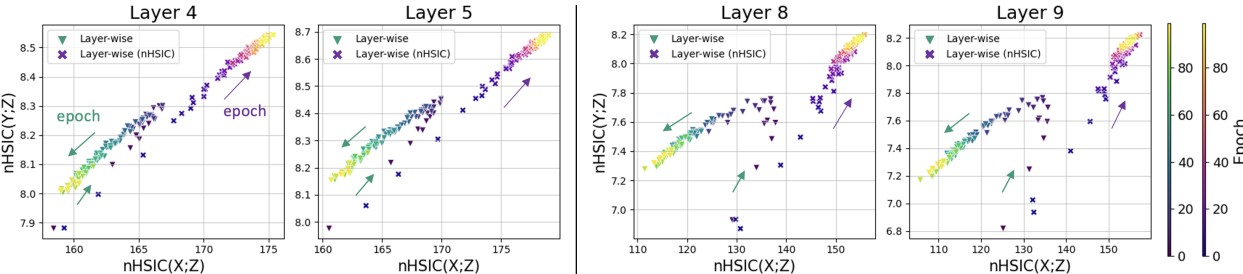

Figure 7: HSIC plane dynamics for layer-wise training with HSIC augmenting term; $\lambda = 0.05$. ResNet18 is trained on CIFAR10, which is the same setting as figure 6. **Left:** Middle layers. **Right:** Output layers. The color gradation shows the progress of training. Inverted triangles with a blue-yellow-based colormap denote layer-wise training in the original setting as a baseline, whereas cross marks with a red-yellow-based colormap show training with HSIC augmenting term.

the training of increasing nHSIC$(X, Z)$, our motivations align with it. This study investigates whether such operations fundamentally address the issues in layer-wise training.

We explore the model performance trained with the local loss with an HSIC augmenting term. Let $\boldsymbol{X} \in \mathbb{R}^{m \times d}$ be $(\boldsymbol{x}_1, \ldots, \boldsymbol{x}_m)^\top$, and $\boldsymbol{Z}_l \in \mathbb{R}^{m \times d_l}$ be the $l$-th layer's output. We adopted the following loss in each layer:

$$\mathcal{L}_{\mathrm{HSIC},l}(\boldsymbol{\theta}_l) = \mathcal{L}_l(\boldsymbol{\theta}_l) - \lambda \cdot \mathrm{nHSIC}(\boldsymbol{X}, f_l(\mathrm{StopGrad}(\boldsymbol{Z}_{l-1}))), \tag{11}$$

where $\lambda > 0$ is a positive hyperparameter to define the regularization strength. In practice, computing the empirical nHSIC for the entire training data requires $\mathcal{O}(m^2)$ memory consumption, so the estimated values within the mini-batch are used for training.

Figure 7 shows the results for layer-wise training with the HSIC augmenting term when $\lambda$ is set to 0.05. While the original layer-wise training exhibited an HSIC decreasing phase, both nHSIC$(X, Z)$ and nHSIC$(Y, Z)$ increased throughout the training without decreasing. Although these HSIC plane dynamics were expected, the actual test accuracy turned out to be worse. Since identity mapping, for example, does not learn useful representations but preserves information, prioritizing information preservation might have hindered the learning of a useful representation here. The results with smaller $\lambda$ are provided in section F.4, but we did not observe any performance improvement from the original setting. Our findings show that 1) the initial HSIC values remained unchanged from the original setting, failing to achieve values as high as those with E2E training, and 2) nHSIC values were uniformly improved in every layer without the middle layers' compression and the information bottleneck behavior in the final layer. The learning dynamics are essentially unchanged from the original layer-wise setting and fail to achieve a cooperative interaction among the layers.

## 5    Compression in Middle Layers from Geometric Perspective

E2E training has an information compression in the middle layers, exhibiting behavior distinct from that of the layers on the output side. However, the benefits of this intermediate compression in E2E training remain a subject of discussion. One reason would be that, based on the previous observations, the compression in the middle layers leads to the information bottleneck behavior in the final layer: a better representation from an information-theoretic perspective. In this section, we discuss the advantages of middle compression from a geometric standpoint. Frosst et al. (2019) stated that entangling representations of different classes in the hidden layers contribute to the final classification performance. Although mutual information does not specify the geometric arrangement of the representation space (Tschannen et al., 2020), we have the following result for the value of HSIC$(Z, Y)$ and *soft nearest neighbor loss* in Frosst et al. (2019).

**Theorem 1** (Informal version of Theorem 4)**.** *Suppose the representation $\boldsymbol{z}$ is bounded as $\|\boldsymbol{z}\|_2 \leq M$ for some constant $M > 0$. If the RBF kernel $k(\boldsymbol{v}, \boldsymbol{w}) = \exp\left(-\|\boldsymbol{v} - \boldsymbol{w}\|_2^2/(2\sigma^2)\right)$ and the linear kernel $l(\boldsymbol{v}, \boldsymbol{w}) = \boldsymbol{v}^\top \boldsymbol{w}$*

*are used for $Z$ and $Y$ respectively, then we have*

$$soft\ nearest\ neighbor\ loss \geq \log c - c \exp\left(\frac{2M^2}{\sigma^2}\right) \mathrm{HSIC}(Y, Z). \tag{12}$$

This theorem implies that reducing $\mathrm{HSIC}(Y, Z)$ leads to an increase in the lower bound of soft nearest neighbor loss, leading to a higher performance of the final layer according to Frosst et al. (2019). Please refer to section E.2 in the appendix for proofs and more detailed interpretation.

## 6 Related Work

**Information and HSIC Bottleneck** Information bottleneck (Tishby et al., 2000; Tishby & Zaslavsky, 2015; Saxe et al., 2018) is one of the concepts for dissecting deep learning. It has been studied in the context of the analysis of the learning dynamics through the information plane (Goldfeld, 2019; Geiger, 2021; Lorenzen et al., 2022; Adilova et al., 2023), the relationship between generalization of neural network (Achille & Soatto, 2018; Chelombiev et al., 2019; Kawaguchi et al., 2023), and the objective of self-supervised representation learning (Sridharan & Kakade, 2008; Federici et al., 2020). The HSIC bottleneck (Ma et al., 2020; Pogodin & Latham, 2020; Wang et al., 2021; Jian et al., 2022; Guo et al., 2023; Wang et al., 2023b) is inspired by the information bottleneck but uses HSIC instead of mutual information because of the simplicity of estimation. In our research, we focus on information plane analysis rather than designing loss functions based on HSIC bottleneck; in this sense, we are influenced by the plots presented in Wang et al. (2021).

**Layer-wise Analysis** While there are prior studies on the differing roles of the layers and compression of representation (Darlow & Storkey, 2020; Peer et al., 2022; Zhang et al., 2022; Chen et al., 2023; Masarczyk et al., 2024), to the best of our knowledge, there has not been any research leveraging the characteristic of layer-wise training, which allows specifying roles for each layer, to reconsider the advantages of E2E training. One study conducted in parallel with ours (Wang et al., 2023a) shares a perspective in analyzing representation learning layer by layer. However, our work differs in that we put emphasis on the information-theoretic interactions between layers for E2E training, while they analyze the geometric representations of classes using deep linear networks.

## 7 Discussion

### 7.1 Forward-Forward Algorithm

One of the merits of layer-wise training is the possibility of adopting a divide-and-conquer approach, which enables us to specify the role of each layer specifically, leading to more interpretable models and parallel efficient training. However, as of this moment, layer-wise training imposes the same objective function on layers, making the role of each layer uniform and negating the importance of stacking layers. We focus on the forward-forward algorithm (Hinton, 2022) as a potential solution to this challenge and investigate its possibility in section C. Unlike the divide-and-conquer approach, it trains layers in a sequential way and forces each layer to learn different features from preceding layers while adopting the same objective function among the layers. This process encourages distinct roles for each layer. While there is a problem in terms of diminishing class information as the layers deepen (see section C), it presents an intriguing approach.

### 7.2 Limitation and Future Direction

In this study, we observed the information compression in the middle layers in E2E training and discussed its benefits from the information bottleneck perspective and the geometric perspective. However, it does not explain why this compression occurs. Furthermore, we do not investigate the merits of backpropagation. The layer-role differentiation observed in E2E training could potentially be obtained by solving the credit assignment problem for each neuron via backpropagation. However, even with biologically plausible methods, if they can address the credit assignment problem appropriately and achieve interaction between layers, there

is a potential to achieve the advantages in E2E training observed in our work. This paper provides criteria that backpropagation alternatives should meet, and exploring this potential is left for future work.

## 8 Conclusion

This study explored the advantages of E2E training through the comparison with layer-wise training. Our information-theoretic analysis using HSIC shows that E2E training propagates efficiently input information and assigns diverse roles to the layers, leading to the information bottleneck representation at the final layer. The information compression of intermediate layers observed in this study could offer an intriguing perspective for understanding neural network behavior in the context of the information bottleneck principle. We leave the investigation to future work of the information compression mechanisms of the middle layers and the influence of layer interaction on the last representation dynamics.

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

# A   Related Work

In the main text, related work for information bottleneck and HSIC-based analysis is presented due to space constraints. Here, we introduce related studies addressing the drawbacks of E2E training and investigating layer-wise training.

**Problems in backpropagation.**   Backpropagation technique (Rumelhart et al., 1986; LeCun et al., 1989) is a foundation of the training methods of current machine learning. While this mechanism has under-pinned recent successes in machine learning, there are still challenges in the following three perspectives: computational problem, biological problem, and interpretability problem (Duan & Principe, 2022).

First, it has computational problems with memory usage and the difficulty of parallel computing. The outputs of intermediate layers are necessary for computing the derivative of the earlier layers; therefore, they should be put on the memory besides the parameter information of the neural network. This can oppress memory usage limitations and lead to the need to compromise the network size and the quality of training data. To alleviate this problem, a check-pointing technique that recomputes the update from checkpoints in the backward step (Chen et al., 2016; Gruslys et al., 2016; Kusumoto et al., 2019) and network compression techniques, such as pruning and quantization, are proposed (Krishnamoorthi, 2018; Blalock et al., 2020; Liang et al., 2021); however, they are still the trade-off between memory usage and training time or test accuracy. In addition, there is an update locking problem, which prevents updating the weights until the forward and backward path of the downstream modules is completely completed, preventing the model parallelism (Krizhevsky, 2014; Huang et al., 2019; Gomez et al., 2022).

Second, training with backpropagation is questioned from a biologically plausible perspective. Biological learning rules are considered to be local in terms of both space and time, and there is a gap from the backpropagation because it satisfies neither of them (Crick, 1989; Baldi et al., 2017). Specifically, if the actual brain adopts the backpropagation-like learning style, we observe various gaps, for example, forward and backward propagation must alternate, the error must be propagated throughout the entire model, the update path must be symmetric because the same weights are used in backward propagation as in forward pass, and continuous values must be represented at the binary value synapses. A Hebbian neural network is one of the techniques to handle the problem of global learning rules (Moraitis et al., 2022; Wang et al., 2022; Journé et al., 2023). Hebbian-like learning methods enable local learning by increasing the weight between presynaptic neurons and postsynaptic neurons when both are activated. Target propagation (Bengio, 2014; Lee et al., 2015; Bartunov et al., 2018) trains each block to regress the target assigned for each block. The key idea of target propagation is backpropagating the target with the approximated inverse of the block instead of backpropagating error with the chain rule of the gradient. This inverted target is generated with the auxiliary networks; therefore, target propagation can alleviate the symmetric weight problem of backpropagation and the problem of gradient instability. A different approach that solves the problem of weight symmetry is feedback alignment (FA) (Lillicrap et al., 2014), which uses the fixed random weight matrix to transport the updated information to the earlier layers. FA alleviated the difficulty of weight symmetry but still has the backward locking problem that the parameter weights cannot be updated until the backward pass has been completed. Direct feedback alignment (DFA) (Nøkland, 2016) passes the gradient information directly to each layer from the output layer so that the weight update can be completed simultaneously.

Finally, training models with backpropagation have a problem of interpretability. The trained network with E2E learning is a black box and difficult to understand its behavior, for example, unexpected performance deterioration is observed in certain situations and the trained model is difficult to be in under control (Goodfellow et al., 2014; Moosavi-Dezfooli et al., 2017). In particular, the intermediate layer's behavior is controlled only through the outputs of downstream layers, and these behaviors are difficult to understand and specify. This fact is one of the reasons that makes understanding the benefits of backpropagation difficult.

**Layer-wise training.**   Layer-wise training was originally studied in the context of getting good initialization to train multi-layer neural networks successfully (Hinton et al., 2006; Bengio et al., 2006). With advancements in the model architecture and the normalization techniques, layer-wise training has gained attention as one of the training methods to address the aforementioned problems with backpropagation. The

classification-based loss is adopted in each layer (Mostafa et al., 2018; Belilovsky et al., 2019; 2020), in contrast, the similarity-based loss is used (Kulkarni & Karande, 2017; Nøkland & Eidnes, 2019; Siddiqui et al., 2023). The essential problem of layer-wise training is that the previous layer cannot use the information of the layer behind it at all. Under the setting that backpropagation within a local block is allowed, Xiong et al. (2020) has tackled this problem so that the information of the subsequent layer can gradually flow to the previous layer by training it by shifting it one layer at a time, using two pairs of layers as one block. Gomez et al. (2022) extends this idea of "overlapping local updates" in a general way, proposing a learning method that balances the high parallelism of layer-wise training with the high prediction accuracy of E2E learning.

Signal propagation (Kohan et al., 2023) has a shared characteristic with similarity-base layer-wise training in that it projects the input and class information into the same representation space and compares them in this space. Associated learning (AL) (Kao & Chen, 2021; Wu et al., 2021) is similar to signal propagation in that it forwards inputs and class labels in the same direction and compares them. The difference with signal propagation is that AL does not share the parameters of the networks used for forwarding inputs and class labels. Recently, the Forward-Forward algorithm has been proposed by Hinton and attracted attention (Hinton, 2022). This learning method shares ideas with signal propagation and AL, but it can enforce the subsequent layers to learn the different features from the previous one. We investigated this possibility in section C in the appendix.

Another direction of this research focuses on the similarity between layer-wise training and ensemble learning (Mosca & Magoulas, 2017; Huang et al., 2018). Huang et al. (2018) proposed the incremental learning methods of ResNet (He et al., 2016) based on research that ResNet can be interpreted as an ensemble of many shallow networks of different lengths (Veit et al., 2016). They also gave a theoretical analysis of this method based on generalization bound for boosting algorithms.

Wang et al. (2020) adds reconstruction error term to the local objective function from the information-theoretic perspective. They hypothesized that layer-wise training aggressively discards the input-related information, which leads to the performance drop from the E2E learning. Our study has been inspired by their work, yet differs in that we employ the information-theoretic tools for analyzing middle layers from the perspective of E2E training, rather than proposing a layer-wise method. Additionally, while their work asserted the effect of increasing $I(X; Z)$ in the layer-wise training, we argue that there still exists a performance gap compared with E2E training in terms of the diversity of roles across layers (see section 4.4).

## B Further Descriptions for Layer-wise Training

### B.1 Details of Training

**Size of local module.** The distinction between the E2E training and layer-wise training allows the continuous middle choice by changing the range granularity of gradient propagation. In various recent studies on local loss training, the entire model is divided into multiple blocks and performs backpropagation within those blocks, instead of strictly decomposed up to the layer level. This approach, which could be termed "block-wise training", can balance the trade-offs between the high accuracy of E2E training and the computational efficiency of layer-wise training, such as memory consumption and parallel computational efficiency. This paper, however, does not aim to propose a practical training method; instead, it primarily focuses on analyzing the reason why E2E training achieves higher accuracy compared to layer-wise training, without considering block-wise training.

**Sequential vs simultaneous training.** In the layer-wise training, there are two major training methods: one sequentially trains layers, adding a new layer on top of fully trained previous modules (Belilovsky et al., 2019; Huang et al., 2018; Marquez et al., 2018); and the other method, for the given model architecture, simultaneously trains each layer based on the local loss of each module in a single forward pass. The sequential training can dynamically decide the model depth depending on the current performance of the validation set; however, it takes more time and achieves lower performance than when each layer is trained simultaneously (Löwe et al., 2019; Wang et al., 2020; Siddiqui et al., 2023). They show that the later modules

have fallen into overfitting, and this can be attributed to the regularizing effect from the noisy input from not fully trained earlier modules in the simultaneous training. Additionally, the main focus of this study lies in its comparison with E2E learning; therefore, we primarily consider the setting of simultaneous training, which forward inputs throughout the entire model as E2E training does. As for the sequential layer-wise training, refer to section F.4.3 in the appendix for experimental results.

## B.2 Layer-wise Training Methods

We conducted experiments in section 2.2 to confirm the performance gap between E2E training and the various layer-wise training methods. In this section, we introduce the loss functions and methods used in the experiments. These methods correspond to the names listed in the table 1 and table 4.

**Cross-Entropy Loss (CE).** The most naive approach is to apply the same cross-entropy loss (CE) as those used in E2E learning. In this case, the class probabilities are calculated using a linear or multi-layer projection head $g_l$ from the intermediate representations, so the local loss is calculated as

$$\mathcal{L}_l(\boldsymbol{\theta}_l) = \frac{1}{m} \sum_{i=1}^{m} \left( -\sum_{k=1}^{c} \mathbf{1}\,(y_i = k) \log \frac{\exp h_{l,i,k}}{\sum_{j=1}^{c} \exp h_{l,i,j}} \right), \tag{13}$$

where $\mathbf{1}$ is an indicator function. Let us remind that the $\boldsymbol{h}_{l,i} \in \mathbb{R}^c$ is the output of $l$-th layer's auxiliary network for input $\boldsymbol{x}_i$, and it is calculated as $\boldsymbol{h}_{l,i} = (g_l \circ f_l)\,(\text{StopGrad}(\boldsymbol{z}_{l-1,i}))$.

**Supervised Contrastive Loss (SupCon).** As a case of optimizing the similarity-base loss at each layer, supervised contrastive (SupCon) loss (Khosla et al., 2020) is used in our experiment. The intermediate representations of an input pair are brought closer together if they belong to the same class and moved apart if they come from different classes. Before training, the data augmentation is performed to assert there is at least one sample of the same class in the batch. Let $N$ be the size of the mini-batch; the new mini-batch of size $2N$ is created with augmented samples $\boldsymbol{x}_{2i-1}, \boldsymbol{x}_{2i}$ from the original $\boldsymbol{x}_i$. In addition, let $P(i) \subset [2N]$ be the set of indices of samples that belong to the same class as $\boldsymbol{x}_i$, except for $i$ itself. The loss is as follows:

$$\mathcal{L}_{l,\text{mini}}(\boldsymbol{\theta}_l) = \frac{1}{2N} \sum_{i=1}^{2N} \frac{-1}{|P(i)|} \sum_{p \in P(i)} \log \frac{\exp\left(\boldsymbol{h}_{l,i}^{\top} \boldsymbol{h}_{l,p}/\tau\right)}{\sum_{a \in [2N] \setminus \{i\}} \exp\left(\boldsymbol{h}_{l,i}^{\top} \boldsymbol{h}_{l,a}/\tau\right)}, \tag{14}$$

where $\tau$ is a hyperparameter for softmax temperature, and $\mathcal{L}_{l,\text{mini}}$ denotes the loss for the mini-batch. In this case, the label information $y$ is not explicitly forwarded, but the class information is given through the other training input belonging to the same class.

**Similarity Matching Loss (Sim).** We use the similarity matching loss from Nøkland & Eidnes (2019) as a similarity-based local loss. While they showed that combining the similarity-based loss and classification-based loss yields a layer-wise trained model with better performance, since the purpose of this study is the analysis of the properties of the trained model, only the similarity-based one is considered. Given outputs after the auxiliary networks $\boldsymbol{H}_l = (\boldsymbol{h}_{l,1}, \ldots, \boldsymbol{h}_{l,m})^{\top} \in \mathbb{R}^{m \times d_l'}$ and the one-hot encoded label information $\boldsymbol{Y}_l = (\boldsymbol{y}_1, \ldots, \boldsymbol{y}_m)^{\top}$, the loss is calculated as:

$$\mathcal{L}_l(\boldsymbol{\theta}_l) = \|\operatorname{sim}(\boldsymbol{H}_l) - \operatorname{sim}(\boldsymbol{Y}_l)\|_F^2, \tag{15}$$

where $\operatorname{sim}(\boldsymbol{H}_l)$ measures the similarities between each rows of $\boldsymbol{H}_l$. Specifically, each element of $\operatorname{sim}(\boldsymbol{H}_l)$ is

$$\operatorname{sim}(\boldsymbol{H}_l)_{i,j} = \frac{\tilde{\boldsymbol{h}}_{l,i}^{\top} \tilde{\boldsymbol{h}}_{l,j}}{\|\tilde{\boldsymbol{h}}_{l,i}\|_2 \|\tilde{\boldsymbol{h}}_{l,j}\|_2}, \tag{16}$$

where $\tilde{\boldsymbol{h}}_{l,i}$ is mean-centered $\boldsymbol{h}_{l,i}$. We calculate $\operatorname{sim}(\boldsymbol{Y}_l)$ in the similar manner.

While both this similarity matching loss and supervised contrastive loss are the similarity-based local loss, one major difference is that the auxiliary network $g$ is composed of one or more convolution layers instead of a linear or MLP (Nøkland & Eidnes, 2019). The output $\boldsymbol{h}$ is obtained by taking the standard deviation for each feature map after the convolutional auxiliary network and concatenating them.

**Signal Propagation (SP).** Signal propagation is one of the layer-wise training methods without relying on the auxiliary networks (Kohan et al., 2023). It trains directly each layer so that the similarities of its outputs would be close to the similarities among the corresponding class labels determined dynamically by the structure of the network. To forward the class information $y$ with the same network structure as the input $\boldsymbol{x}$, an auxiliary network is used to map $y$ to the same input space as $\boldsymbol{x}$. This network is required only at the network input, which is different from the auxiliary networks in other layer-wise methods, so the parameter size becomes small.

We denote by $\boldsymbol{S}_l = (\boldsymbol{s}_{l,1}, \ldots, \boldsymbol{s}_{l,m})^\top \in \mathbb{R}^{m \times d_l}$ the class information of each example forwarded until the $l$-th layer, which is called signal. Given the intermediate representations $\boldsymbol{Z}_l = (\boldsymbol{z}_{l,1}, \ldots, \boldsymbol{z}_{l,m})^\top \in \mathbb{R}^{m \times d_l}$, we train $f_l$ so that the similarities between the elements of $\boldsymbol{S}_{l-1}$ and $\boldsymbol{Z}_l$ is close. That is, each layer is trained so that the layer's outputs would be in the spacial relationships based on the similarity of the class information forwarded up to that layer. The similarities among the representations $\boldsymbol{Z}_l$ are calculated by dot product and row-wise softmax function, like the attention matrix calculation in the transformer (Vaswani et al., 2017). The loss for signal propagation is specifically calculated as follows:

$$\mathcal{L}_l(\boldsymbol{\theta}_l) = \frac{1}{m} \sum_{i=1}^m \sum_{j=1}^m \left( -\left(\mathrm{sim}_s\left(\boldsymbol{S}_{l-1}\right)\right)_{i,j} \log\left(\mathrm{sim}_z\left(\boldsymbol{Z}_l\right)\right)_{i,j} \right), \tag{17}$$

where

$$\mathrm{sim}_z\left(\boldsymbol{Z}_l\right) = \mathrm{RowWiseSoftmax}\left(\frac{\boldsymbol{Z}_l \boldsymbol{Z}_l^\top}{d_l}\right). \tag{18}$$

There can be two types of the $\mathrm{sim}_s$ settings. In the original implementation (Kohan et al., 2023), the top several, e.g. six, examples with the highest similarity are set to one, and the rest are set to zero for each example. Another naive choice is to apply row-wise softmax normalization similar to $\mathrm{sim}_z$. The former is denoted as "SP (hard)" and the latter as "SP (soft)". This loss can also be thought of as optimizing the Kullback-Leibler divergence between the empirical distributions of class information's similarity and the output similarity for each example.

## C  Forward-Forward Algorithm

In this section, we investigate the Forward-Forward algorithm, which is one of the layer-wise training methods, proposed by Hinton (2022). In this algorithm, the input $\boldsymbol{x}$ is combined with label information $y$ using techniques like one-hot encoding. This combined input is passed through the network, where each layer is trained to distinguish whether the input pair $(\boldsymbol{x}, y)$ is a correct label pair or not. Specifically, when the input pair $(\boldsymbol{x}, y)$ is correctly labeled, the L2 norm of the layer's output is enlarged; conversely, if it is incorrectly labeled, the norm is diminished.

The key point of this method is the normalization of the outputs at each layer. Each layer trains the similarity itself of the input pair through its output norm and normalizes the output's L2 norm before passing it to the subsequent layer. This normalization aligns with the norms of the representations; therefore, in the training of the subsequent layers, the features learned in the previous layer cannot be utilized. As a result, stacking layers takes on explicit significance in extracting complex information about the input pair similarity. In the case of layer-wise training methods based on the spacial relationship in the representation space, even if a layer has a meaningless parameter, e.g. an identity mapping, the performance does not deteriorate as the representation from the previous layer can be directly reused. However, **the forward-forward algorithm constantly imposes a new role for feature extraction on each layer**, potentially enabling the diversity of each layer's role, similar to the advantage of E2E training discussed in the main text. This is why this algorithm is specifically addressed in this section.

An inherent issue arises in the original setting, where passing through more convolutional layers leads to the loss of class information. This is because one-hot encoded information can be easily lost by the convolutional operation.

To address this, we explored two modifications.

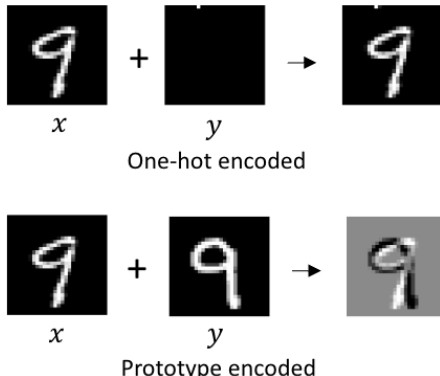

Figure 8: Possible methods for embedding label information in the forward-forward algorithm. **Top:** The label is one-hot encoded to the top-left of images. **Bottom:** The actual image that represents the label, i.e. prototype, is combined with the input.

Table 3: Test accuracy of the original forward-forward algorithm and modified version. For MNIST training, a toy MLP model that has four layers is used, while a toy model with two convolutional layers and two linear layers is used for CIFAR10.

|          | MNIST | CIFAR10 |
|----------|-------|---------|
| Original | 96.4  | 9.4     |
| Ours     | **98.0** | **48.7** |

1. Use class prototypes instead of one-hot encoding to preserve the label information through convolution. Figure 8 illustrates this change. These prototypes could be fixed images for each class, randomly selected images from each class, or the average images within each class.

2. Change the location of label information mixing. In the original setting, the class information is only mixed before the entire model. We revised this approach to mix class information just before the currently trained layer and forward $x$ and $y$ separately through the model to mitigate the loss of class information.

Table 3 shows the test accuracy for the MNIST dataset and CIFAR10 dataset when using the original forward-forward algorithm and our modified version. The test accuracy is improved from the original algorithm, and particularly for CIFAR10, the issue that the model cannot be trained with the forward-forward algorithm has been resolved. Although this modification has led to improved accuracy to some extent, there is still a performance gap compared to other existing layer-wise training methods. Moreover, despite expecting that the forward-forward algorithm could extract more complex features as the layers are stacked, the accuracy deteriorates, rather than improves. These results suggest that despite these modifications to prevent loss of the class information, the information about inputs or class labels is being lost or inadequately propagated as the network deepens.

# D  Mutual Information and HSIC

## D.1  Mutual Information

The concept of assessing the amount of information for one random variable by observing the other one is mutual information. The mutual information between two random variables $X$ and $Y$ is defined as,

$$I(X;Y) := \int \int p_{XY}(x,y) \log \frac{p_{XY}(x,y)}{p_X(x)p_Y(y)} dx dy,$$

where $p_{XY}$ is the density functions of the joint distribution, and $p_X$ and $p_Y$ are the density functions of the marginal distributions. It equals the KL divergence between a joint distribution and the product of marginal distributions. In the field of machine learning, mutual information has been widely employed in contexts such as representation learning and feature selection (Estévez et al., 2009; Beraha et al., 2019; Hjelm et al., 2018; Tian et al., 2020).

### D.2 Difficulty in Neural Network Analysis based on Mutual Information

Analyzing mutual information within the deep neural network is an intriguing research subject; however, it faces two challenges, as stated below.

The first one is that mutual information can be infinite under certain formulations, which makes the comparison of mutual information meaningless. We naturally consider that the input $X$ and intermediate representations $Z$ are continuous random variables in a real vector space. However, if $Z$ is determined by $X$ as $Z = f(X)$ in a deterministic way, and if the entropy of $Z$ is finite, the conditional density distribution turns into a Dirac delta function, leading to infinite mutual information. Amjad & Geiger (2019) demonstrates that $I(X; Z)$ becomes infinite under certain assumptions about the activation function. One of the methods to avoid this problem is adding noise to the middle representation, but it does not align with the actual behavior of the network. Note that the actual $Z$ is a discrete variable because of the precision of the floating-point, but estimating the probability of falling into all bins is intractable.

The second problem is the difficulty of estimating mutual information. Measuring mutual information is an intrinsically difficult problem due to the high dimensionality and density ratio. The initial attempt to analyze the DNN from the IB perspective used the binning-based estimator for measuring mutual information; however, it does not scale to high-dimensional settings and the IB phenomenon was not observed for the unbounded ReLU activations, which is commonly used in the machine learning implementations. Other approaches to estimating mutual information are a non-parametric KDE estimator and a k-NN-based estimator. Using these algorithms still suffers from the high dimensionality in the network representations.

### D.3 HSIC and Chi-Squared Mutual Information

To our knowledge, there is no clear relationship between mutual information and HSIC. In terms of the independence of two random variables $X, Y$, $I(X; Y) = 0$ if and only if $X$ and $Y$ are independent. The independence of $X$ and $Y$ implies $\text{HSIC}(X, Y) = 0$, and other direction holds when the product kernel $kl$ is *characteristic* (Fukumizu et al., 2007; Sriperumbudur et al., 2011), i.e., the probability distribution on the input space corresponds one-to-one with the mean in the RKHS. This property does not hold with a linear and polynomial kernel, however, the Gaussian RBF kernel used in this study is characteristic.

Moreover, besides independence, this characteristic kernel gives a further interpretation of HSIC values. Fukumizu et al. (2007) demonstrated that the normalized HSIC value equals the $\chi_2$-divergence between the joint distribution and the product of the marginal distributions, which is a variant of mutual information. Let us review the definition of the $\chi_2$-divergence and chi-squared mutual information.

**Definition 1** ($\chi_2$-divergence). *Let $P_X$ and $Q_X$ be the distributions of random variable $X$, with the probability density function $p_X(\mathrm{x})$ and $q_X(\mathrm{x})$, respectively. The $\chi_2$-divergence between these two distributions is*

$$\chi_2(P_X \| Q_X) = \int \int \left( \frac{p_X(x)}{q_X(x)} - 1 \right)^2 q_X(x) dx. \tag{19}$$

**Definition 2** (chi-squared mutual information). *Let $X$ and $Y$ be the two random variables. If their joint distribution is $P_{XY}$ and the marginal distributions are $P_X$ and $P_Y$, with the probability density function $p_{XY}(x, y)$, $p_X(x)$, and $p_Y(y)$, the chi-squared mutual information between these two random variables is defined by*

$$I_{\chi_2}(X; Y) = \int \int \left( \frac{p_{XY}(x, y)}{p_X(x) p_Y(y)} - 1 \right)^2 p_X(x) p_Y(y) dx dy. \tag{20}$$

Mutual information is defined as the KL divergence between the joint distribution and the marginal distributions, and the used divergence is different from the chi-squared mutual information. Both of these distances for measures belong to the class known as $f$-divergence. These values cannot be directly computed from the other value, but the following inequality relationship holds.

**Theorem 2** (Gibbs & Su (2002), Theorem 5). *The KL divergence and $\chi_2$-divergence satisfy*

$$\mathrm{KL}(P_X \| Q_X) \leq \log\left[1 + \chi_2(P_X \| Q_X)\right]. \tag{21}$$

From this inequality, we can also obtain the relationship for mutual information:

$$I(X;Y) \leq \log\left[1 + I_{\chi_2}(X;Y)\right] \leq I_{\chi_2}(X;Y). \tag{22}$$

We introduced chi-squared mutual information and discussed its relationship with the standard mutual information. Next, we give the result that normalized HSIC equals this chi-squared mutual information.

**Theorem 3** (Fukumizu et al. (2007), Theorem 4). *Let $X$ and $Y$ be the two random variables on $\mathcal{X}$ and $\mathcal{Y}$, and let $\mathcal{H}_{\kappa_1}$ and $\mathcal{H}_{\kappa_2}$ be the associated RKHSs with the kernels $\kappa_1 : \mathcal{X} \times \mathcal{X} \to \mathbb{R}$ and $\kappa_2 : \mathcal{Y} \times \mathcal{Y} \to \mathbb{R}$, respectively. Assume that there exists probability density functions $p_{XY}$, $p_X$, and $p_Y$ for the joint distribution and the marginal distributions. If $(\mathcal{H}_{\kappa_1} \otimes \mathcal{H}_{\kappa_2}) + \mathbb{R}$ is dense in $L^2(P_X \otimes P_Y)$, and the Hilbert-Schmidt norm of the operator $\mathcal{C}_{XX}^{-1/2} \mathcal{C}_{XY} \mathcal{C}_{YY}^{-1/2}$ exists, then we have*

$$\mathrm{nHSIC}(X,Y) = \|\mathcal{C}_{XX}^{-1/2} \mathcal{C}_{XY} \mathcal{C}_{YY}^{-1/2}\|_{HS}^2 = I_{\chi_2}(X;Y). \tag{23}$$

According to Proposition 5 in Fukumizu et al. (2009), the characteristic property of the kernel and the condition part of this theorem, where $(\mathcal{H}_{\kappa_1} \otimes \mathcal{H}_{\kappa_2}) + \mathbb{R}$ is dense in $L^2(P_X \otimes P_Y)$, are equivalent. Therefore, if the product kernel $kl$ is characteristic, then nHSIC equals the chi-squared mutual information. This theorem justifies the use of normalized HSIC to analyze the information flow within the neural network model.

When considering continuous random variables for input $X$ and representation $Z$, we can use the RBF kernel and this theorem. In contrast, since the label $Y$ is a discrete random variable, which does not have a density function, it necessitates a re-definition of the mutual information. While the relationship between nHSIC and chi-squared mutual information in this setting is not explicit, it can be verified that the condition similar to that in this theorem is satisfied when a linear kernel is applied to label $Y$.

**Proposition 1.** *Let $Y$ be a random variable that takes values on the one-hot encoded classes $\mathcal{Y} = \{e_1, \ldots, e_c\}$, and $\mathcal{H}$ is an associated RKHS with a linear kernel $\kappa$ on $\mathbb{R}^c \times \mathbb{R}^c$. For the two probability distributions $P$ and $Q$, if $\mathbb{E}_P[f(Y)] = E_Q[f(Y)]$ for any $f \in \mathcal{H}$, then $P = Q$.*

*Proof.* Use the condition for $e_i \in \mathcal{H} = \{x \mapsto v^\top x | v \in \mathbb{R}^c\}$, leading to $P(Y = e_i) = \mathbb{E}_P[e_i^\top Y] = \mathbb{E}_Q[e_i^\top Y] = Q(Y = e_i)$. The same holds for $e_1, \ldots, e_c$, and we get $P = Q$. $\qquad\square$

# E   Further Details

## E.1   Lower Bound of Mutual Information

In the main text, we briefly mentioned the relationship between the lower bound of mutual information and the optimization of cross-entropy loss in equation 8. Here, we discuss this relationship in a bit more detail.

The non-negativity of KL divergence yields the following lower bound on mutual information (Barber & Agakov, 2004).

$$I(Z;Y) \geq \mathbb{E}_{p(y,z)}\left[\log q(y|z)\right] + H(Y), \tag{24}$$

where the equality holds when $q(y|z)$ equals to $p(y|z)$.

The initial term becomes

$$\mathbb{E}_{p(y,\boldsymbol{z})}\left[\log q(y|\boldsymbol{z})\right] = \int dy d\boldsymbol{z} \, p(y, \boldsymbol{z}) \log q(y|\boldsymbol{z}) \tag{25}$$

$$= \int dy d\boldsymbol{z} d\boldsymbol{x} \, p(y|\boldsymbol{x}) p(\boldsymbol{z}|\boldsymbol{x}) p(\boldsymbol{x}) \log q(y|\boldsymbol{z}) \tag{26}$$

$$= \int dy d\boldsymbol{x} \, p(y, \boldsymbol{x}) \int d\boldsymbol{z} \, p(\boldsymbol{z}|\boldsymbol{x}) \log q(y|\boldsymbol{z}) \tag{27}$$

$$= \mathbb{E}_{p(y,\boldsymbol{x})}\left[\mathbb{E}_{p(\boldsymbol{z}|\boldsymbol{x})}\left[\log q(y|\boldsymbol{z})\right]\right]. \tag{28}$$

In equation 26, we use $p(y, \boldsymbol{z}|\boldsymbol{x}) = p(y|\boldsymbol{x})p(\boldsymbol{z}|\boldsymbol{x})$ derived from the Markov chain $Y \to X \to Z$. This takes a similar form to the negative cross-entropy loss. In fact, the population form of cross-entropy loss in equation 13 is as follows:

$$\mathcal{L}_{CE} = -\mathbb{E}_{p(y,\boldsymbol{x})}\left[\log \frac{\exp \hat{y}_y}{\sum_{j=1}^{c} \exp \hat{y}_j}\right], \tag{29}$$

where $\hat{\boldsymbol{y}}$ is the output of the classifier, in this case, the auxiliary classifier's output $\boldsymbol{h}$. We are now considering the case where $\boldsymbol{z}$ is determined by $\boldsymbol{x}$; therefore, $\boldsymbol{z}$ can be expressed as $\boldsymbol{z} = f(\boldsymbol{x})$, where $f$ is the backbone model up to the target layer. At this time, $\mathbb{E}_{p(\boldsymbol{z}|\boldsymbol{x})}\left[\log q(y|\boldsymbol{z})\right] = \log q(y|f(\boldsymbol{x}))$ holds in equation 28. If we represent the variational distribution $q$ by the output of the auxiliary classifier $g$ and softmax function, this value corresponds to the cross entropy-loss equation 29; consequently, minimizing the cross-entropy loss implies maximizing the lower bound in equation 8.

## E.2 Soft Nearest Neighbor Loss and HSIC

In this section, we highlight the relationship between the HSIC values of the intermediate representation and the soft nearest neighbor loss (Salakhutdinov & Hinton, 2007; Frosst et al., 2019). We consider the advantages of the middle layer's compression in E2E training through the connection with soft nearest neighbor loss. Note that while nHSIC is used in the information plane plot, the discussion here is based on HSIC.

**Definition 3** (soft nearest neighbor loss, original form Frosst et al. (2019)). *The soft nearest neighbor loss at temperature $T$, for a batch of $b$ samples $(x, y)$ is*

$$\ell'_{sn}(x, y, T) := -\frac{1}{b}\sum_{i \in 1,\dots,b} \log \left(\frac{\displaystyle\sum_{\substack{j \in 1,\dots,b \\ j \neq i \\ y_i = y_j}} \exp\left(-\frac{\|x_i - x_j\|_2^2}{T}\right)}{\displaystyle\sum_{\substack{k \in 1,\dots,b \\ k \neq i}} \exp\left(-\frac{\|x_i - x_k\|_2^2}{T}\right)}\right), \tag{30}$$

*where $x$ can be either the raw input or its representation in some hidden layer.*

Frosst et al. (2019) analyze the structure of middle representations through this value. They argue that **maximizing the entanglement of representations**, i.e., maximizing soft nearest neighbor loss, is beneficial for the discrimination in the last layer. In the main text, we observed the compression phase of HSIC value in the middle layers even with the E2E training. Although reducing $\text{nHSIC}(X, Z)$ in the middle layers is understandable because it leads to the decrease in the final layer's $\text{nHSIC}(X, Z)$, reducing $\text{nHSIC}(Y, Z)$ is intriguing as it reduces the necessary information for final prediction. The main theorem in this section suggests that minimizing the value of $\text{HSIC}(Y, Z)$ contributes to increasing soft nearest neighbor loss; therefore, based on Frosst et al. (2019), it promotes the entanglement of representation and leads to the final better performance.

In this section, we consider the population version of soft nearest neighbor loss and the intermediate representation for $x$; we use $\boldsymbol{z}$. We sample $n$ instances from the conditional distribution of the class instead of counting the examples that belong to the same class for a fixed batch. The dataset is assumed to be class-balanced in this analysis. We define the soft nearest neighbor loss as follows.

**Definition 4** (soft nearest neighbor loss). *For a class-balanced dataset with $c$ classes, the soft nearest neighbor loss at temperature at $T$ is defined as:*

$$\ell_{sn}(n, T) := - \mathop{\mathbb{E}}_{\substack{(y,\boldsymbol{z}) \sim p(y,\boldsymbol{z}) \\ \{\boldsymbol{z}_i^+\}_{i=1}^n \overset{i.i.d.}{\sim} p(\boldsymbol{z}|y) \\ \{\boldsymbol{z}_i^-\}_{i=1}^{cn} \overset{i.i.d.}{\sim} p(\boldsymbol{z})}} \log \left( \frac{\sum_{i=1}^n \exp\left(-\frac{\|\boldsymbol{z} - \boldsymbol{z}_i^+\|_2^2}{T}\right)}{\sum_{i=1}^{cn} \exp\left(-\frac{\|\boldsymbol{z} - \boldsymbol{z}_i^-\|_2^2}{T}\right)} \right). \tag{31}$$

Please note that since the examples from the numerator do not appear in the denominator summation, the loss is not necessarily non-negative.

When the number of positive samples $n$ goes to infinity, and consequently the total number of data $cn$ also goes to infinity, such that the inside of the logarithm converges to the expected value, the following result is obtained regarding the soft nearest neighbor loss and HSIC.

**Theorem 4** (Formal version of Theorem 1.). *Suppose the representation $\boldsymbol{z}$ is bounded as $\|\boldsymbol{z}\|_2 \leq M$ for some constant $M > 0$. The RBF kernel $k(\boldsymbol{v}, \boldsymbol{w}) = \exp\left(-\|\boldsymbol{v} - \boldsymbol{w}\|_2^2/(2\sigma^2)\right)$ and the linear kernel $l(\boldsymbol{v}, \boldsymbol{w}) = \boldsymbol{v}^\top \boldsymbol{w}$ are used for $Z$ and $Y$ respectively. As $n$ goes to infinity, the left-hand side of the following equation converges, and we have*

$$\lim_{n \to \infty} \ell_{sn}(n, 2\sigma^2) \geq \log c - c \exp\left(\frac{2M^2}{\sigma^2}\right) \mathrm{HSIC}(Y, Z). \tag{32}$$

This result might raise concerns about providing a vacuous bound of the loss. However, we can observe the following three facts: 1) $\ell_{sn}$ is not necessarily non-negative as mentioned earlier, 2) the loss value becomes $\log c$ even in the poor representation where all vectors collapse to a single point, and 3) $\mathrm{HSIC}(Y, Z)$ remains at most $1/c$, as indicated by the following lemma 1. These facts allow the meaningful bound by appropriately selecting the value of $\sigma$. In the nHSIC analysis in this study, $\sigma$ is chosen so that $\sigma^2$ would be proportional to the dimensions of the representation space, resulting in the small values of HSIC itself.

We start with the following two lemmas before presenting the proof.

**Lemma 1.** *Suppose that the label $y$ is one-hot encoded to $\boldsymbol{y} \in \{0, 1\}^c$, and the linear kernel $l(\boldsymbol{v}, \boldsymbol{w}) = \boldsymbol{v}^\top \boldsymbol{w}$ is used for $Y$. Then, the exact form of $\mathrm{HSIC}(Y, Z)$ is*

$$\mathrm{HSIC}(Y, Z) = \frac{1}{c} \left( \mathbb{E}_{p_{pos}(\boldsymbol{z}, \boldsymbol{z}')} [k(\boldsymbol{z}, \boldsymbol{z}')] - \mathbb{E}_{p(\boldsymbol{z})p(\boldsymbol{z}')} [k(\boldsymbol{z}, \boldsymbol{z}')] \right), \tag{33}$$

*where $p_{pos}(\boldsymbol{z}, \boldsymbol{z}') = \sum_{y=1}^c p(y)p(\boldsymbol{z}|y)p(\boldsymbol{z}'|y)$.*

This is a supervised setting of Theorem A.1 in Li et al. (2021).

*Proof.* From equation 7,

$$\mathrm{HSIC}(Y, Z) = \mathbb{E}_{p(\boldsymbol{y}, \boldsymbol{z})p(\boldsymbol{y}', \boldsymbol{z}')}[k(\boldsymbol{z}, \boldsymbol{z}')l(\boldsymbol{y}, \boldsymbol{y}')] - 2\mathbb{E}_{p(\boldsymbol{y}, \boldsymbol{z})}\left[\mathbb{E}_{p(\boldsymbol{z}')}[k(\boldsymbol{z}, \boldsymbol{z}')|\boldsymbol{z}]\mathbb{E}_{p(\boldsymbol{y})}[l(\boldsymbol{y}, \boldsymbol{y}')|\boldsymbol{y}]\right]$$
$$+ \mathbb{E}_{p(\boldsymbol{z})p(\boldsymbol{z}')}[k(\boldsymbol{z}, \boldsymbol{z}')]\mathbb{E}_{p(\boldsymbol{y})p(\boldsymbol{y}')}[l(\boldsymbol{y}, \boldsymbol{y}')]. \tag{34}$$

Since the linear kernel $l$ is applied to the one-hot vector $\boldsymbol{y}$,

$$l(\boldsymbol{y}, \boldsymbol{y}') = \begin{cases} 1 & (y = y') \\ 0 & (\text{otherwise}) \end{cases} = \mathbf{1}(y = y'). \tag{35}$$

Using this, the first term of equation 34 is

$$\mathbb{E}_{p(\boldsymbol{y},\boldsymbol{z})p(\boldsymbol{y}',\boldsymbol{z}')}[k(\boldsymbol{z},\boldsymbol{z}')l(\boldsymbol{y},\boldsymbol{y}')] = \frac{1}{c^2}\sum_{y=1}^{c}\sum_{y'=1}^{c}\mathbb{E}_{p(\boldsymbol{z}|y)p(\boldsymbol{z}'|y')}\left[k(\boldsymbol{z},\boldsymbol{z}')\mathbf{1}(y=y')\right] \tag{36}$$

$$= \frac{1}{c^2}\sum_{y=1}^{c}\mathbb{E}_{p(\boldsymbol{z}|y)p(\boldsymbol{z}'|y)}\left[k(\boldsymbol{z},\boldsymbol{z}')\right] \tag{37}$$

$$= \frac{1}{c}\mathbb{E}_{p_{pos}(\boldsymbol{z},\boldsymbol{z}')}[k(\boldsymbol{z},\boldsymbol{z}')]. \tag{38}$$

In the last line, we used $p(y) = 1/c, y \in \{1,\dots,c\}$ and the definition of $p_{pos}$.

The second term is written as

$$\mathbb{E}_{p(\boldsymbol{y},\boldsymbol{z})}\left[\mathbb{E}_{p(\boldsymbol{z}')}[k(\boldsymbol{z},\boldsymbol{z}')|\boldsymbol{z}]\mathbb{E}_{p(\boldsymbol{y})}[l(\boldsymbol{y},\boldsymbol{y}')|\boldsymbol{y}]\right] = \mathbb{E}_{p(\boldsymbol{y},\boldsymbol{z})}\left[\mathbb{E}_{p(\boldsymbol{z}')}[k(\boldsymbol{z},\boldsymbol{z}')|\boldsymbol{z}]\frac{1}{c}\right] \tag{39}$$

$$= \frac{1}{c}\mathbb{E}_{p(\boldsymbol{z})}\left[\mathbb{E}_{p(\boldsymbol{z}')}[k(\boldsymbol{z},\boldsymbol{z}')|\boldsymbol{z}]\right] \tag{40}$$

$$= \frac{1}{c}\mathbb{E}_{p(\boldsymbol{z})p(\boldsymbol{z}')}\left[k(\boldsymbol{z},\boldsymbol{z}')\right]. \tag{41}$$

Finally, the third term is

$$\mathbb{E}_{p(\boldsymbol{z})p(\boldsymbol{z}')}[k(\boldsymbol{z},\boldsymbol{z}')]\mathbb{E}_{p(\boldsymbol{y})p(\boldsymbol{y}')}[l(\boldsymbol{y},\boldsymbol{y}')] = \frac{1}{c}\mathbb{E}_{p(\boldsymbol{z})p(\boldsymbol{z}')}[k(\boldsymbol{z},\boldsymbol{z}')]. \tag{42}$$

Combining these yields the conclusion of the lemma. $\qquad\square$

**Lemma 2.** *Let $s,t$ be the real number satisfying $r \leq s \leq t \leq 1$, where $r > 0$ is the constant. We have*

$$\log t - \log s \leq \frac{1}{r}(t-s). \tag{43}$$

*Proof.* If $s < t$ holds, from the mean value theorem, there exists $c$ in $(s,t)$ such that

$$\log t - \log s = \frac{1}{c}(t-s). \tag{44}$$

Since $r < c$, we get $\log t - \log s < \frac{1}{r}(t-s)$. In the case of $t = s$, both sides of the inequality are equal to zero, thus proving the lemma's statement. $\qquad\square$

Next, we prove the theorem 4 using these lemmas.

*Proof of theorem 4.* We first show the limit of the left-hand side exists.

$$\ell_{sn}(n,2\sigma^2) = -\mathop{\mathbb{E}}_{\substack{(y,\boldsymbol{z})\sim p(y,\boldsymbol{z}) \\ \{\boldsymbol{z}_i^+\}_{i=1}^{n} \overset{i.i.d.}{\sim} p(\boldsymbol{z}|y) \\ \{\boldsymbol{z}_i^-\}_{i=1}^{cn} \overset{i.i.d.}{\sim} p(\boldsymbol{z})}} \log\left(\frac{n\cdot\frac{1}{n}\sum_{i=1}^{n}k(\boldsymbol{z},\boldsymbol{z}_i^+)}{cn\cdot\frac{1}{cn}\sum_{i=1}^{cn}k(\boldsymbol{z},\boldsymbol{z}_i^-)}\right) \tag{45}$$

$$= \log c - \mathop{\mathbb{E}}_{\substack{(y,\boldsymbol{z})\sim p(y,\boldsymbol{z}) \\ \{\boldsymbol{z}_i^+\}_{i=1}^{n} \overset{i.i.d.}{\sim} p(\boldsymbol{z}|y)}} \log\left(\frac{1}{n}\sum_{i=1}^{n}k(\boldsymbol{z},\boldsymbol{z}_i^+)\right) + \mathop{\mathbb{E}}_{\substack{(y,\boldsymbol{z})\sim p(y,\boldsymbol{z}) \\ \{\boldsymbol{z}_i^-\}_{i=1}^{cn} \overset{i.i.d.}{\sim} p(\boldsymbol{z})}} \log\left(\frac{1}{cn}\sum_{i=1}^{cn}k(\boldsymbol{z},\boldsymbol{z}_i^-)\right). \tag{46}$$

Regarding the inside of the expected value in the second term, using the strong law of large numbers and the continuous mapping theorem from the continuity of the log function, we obtain the following:

$$\lim_{n\to\infty}\log\left(\frac{1}{n}\sum_{i=1}^{n}k(\boldsymbol{z},\boldsymbol{z}_i^+)\right) = \log\left(\mathop{\mathbb{E}}_{\boldsymbol{z}^+\sim p(\boldsymbol{z}|y)}\left[k(\boldsymbol{z},\boldsymbol{z}^+)\right]\right) \quad a.s.\,. \tag{47}$$

From this convergence of almost everywhere and the fact that $\left|\log(\frac{1}{n}\sum_{i=1}^{n} k(\boldsymbol{z}, \boldsymbol{z}_i^+))\right|$ is bounded for any $n = 1, 2, \ldots$, which is derived from the assumption $\|\boldsymbol{z}\|_2 \leq M$ for any $\boldsymbol{z}$, the dominated convergence theorem yields

$$\lim_{n\to\infty} \mathbb{E}_{\substack{(y,\boldsymbol{z})\sim p(y,\boldsymbol{z}) \\ \{\boldsymbol{z}_i^+\}_{i=1}^n \overset{i.i.d.}{\sim} p(\boldsymbol{z}|y)}} \log\left(\frac{1}{n}\sum_{i=1}^{n} k(\boldsymbol{z}, \boldsymbol{z}_i^+)\right) = \mathbb{E}_{\substack{(y,\boldsymbol{z})\sim p(y,\boldsymbol{z}) \\ \{\boldsymbol{z}_i^+\}_{i=1}^n \overset{i.i.d.}{\sim} p(\boldsymbol{z}|y)}} \lim_{n\to\infty} \log\left(\frac{1}{n}\sum_{i=1}^{n} k(\boldsymbol{z}, \boldsymbol{z}_i^+)\right) \tag{48}$$

$$= \mathbb{E}_{(y,\boldsymbol{z})\sim p(y,\boldsymbol{z})} \log\left(\mathbb{E}_{\boldsymbol{z}^+\sim p(\boldsymbol{z}|y)}\left[k(\boldsymbol{z}, \boldsymbol{z}^+)\right]\right). \tag{49}$$

Applying a similar reasoning to the third term of equation 46,

$$\lim_{n\to\infty} \mathbb{E}_{\substack{(y,\boldsymbol{z})\sim p(y,\boldsymbol{z}) \\ \{\boldsymbol{z}_i^-\}_{i=1}^{cn} \overset{i.i.d.}{\sim} p(\boldsymbol{z})}} \log\left(\frac{1}{cn}\sum_{i=1}^{cn} k(\boldsymbol{z}, \boldsymbol{z}_i^-)\right) = \mathbb{E}_{(y,\boldsymbol{z})\sim p(y,\boldsymbol{z})} \log\left(\mathbb{E}_{\boldsymbol{z}^-\sim p(\boldsymbol{z})}\left[k(\boldsymbol{z}, \boldsymbol{z}^-)\right]\right). \tag{50}$$

Taking the limit of both side of equation 46 leads

$$\lim_{n\to\infty} \ell_{sn}(n, 2\sigma^2) = \log c - \mathbb{E}_{p(y,\boldsymbol{z})} \log\left(\mathbb{E}_{p(\boldsymbol{z}'|y)}[k(\boldsymbol{z}, \boldsymbol{z}')]\right) + \mathbb{E}_{p(y,\boldsymbol{z})} \log\left(\mathbb{E}_{p(\boldsymbol{z}')}[k(\boldsymbol{z}, \boldsymbol{z}')]\right) \tag{51}$$

In the remainder, we proceed to get the lower bound related to HSIC.

$$\lim_{n\to\infty} \ell_{sn}(n, 2\sigma^2) = \log c - \sum_{y=1}^{c} \int d\boldsymbol{z} p(\boldsymbol{z}|y) \log\left(\int d\boldsymbol{z}' p(\boldsymbol{z}'|y) k(\boldsymbol{z}, \boldsymbol{z}')\right) + \int d\boldsymbol{z} p(\boldsymbol{z}) \log\left(\int d\boldsymbol{z}' p(\boldsymbol{z}') k(\boldsymbol{z}, \boldsymbol{z}')\right) \tag{52}$$

$$\geq \log c - \log\left(\mathbb{E}_{p_{pos}(\boldsymbol{z},\boldsymbol{z}')}[k(\boldsymbol{z}, \boldsymbol{z}')]\right) + \int d\boldsymbol{z} p(\boldsymbol{z}) \log\left(\int d\boldsymbol{z}' p(\boldsymbol{z}') k(\boldsymbol{z}, \boldsymbol{z}')\right) \tag{53}$$

$$= \log c - \int d\boldsymbol{z} p(\boldsymbol{z})\left(\log\left(\mathbb{E}_{p_{pos}(\boldsymbol{z},\boldsymbol{z}')}[k(\boldsymbol{z}, \boldsymbol{z}')]\right) - \log\left(\int d\boldsymbol{z}' p(\boldsymbol{z}') k(\boldsymbol{z}, \boldsymbol{z}')\right)\right). \tag{54}$$

Equation 53 follows from Jensen's inequality.

We apply the lemma 2 to lower-bound the difference in logs. As the inside of the exponential in kernel $k$ is non-negative, it is less than or equal to 1. Given the assumption on the norm of the intermediate representation, $\|z\|_2 \leq M$, the value of kernel becomes greater than or equal to $\exp(-(2M^2)/\sigma^2)$. Since we have $\exp(-(2M^2)/\sigma^2) \leq k(\boldsymbol{z}, \boldsymbol{z}') \leq 1$ for any $\boldsymbol{z}, \boldsymbol{z}'$, we can further lower-bound it using lemma 2 as follows:

$$\lim_{n\to\infty} \ell_{sn}(n, 2\sigma^2) \geq \log c - \int d\boldsymbol{z} p(\boldsymbol{z})\left(\exp\left(\frac{2M^2}{\sigma^2}\right)\left(\mathbb{E}_{p_{pos}(\boldsymbol{z},\boldsymbol{z}')}[k(\boldsymbol{z}, \boldsymbol{z}')] - \int d\boldsymbol{z}' p(\boldsymbol{z}') k(\boldsymbol{z}, \boldsymbol{z}')\right)\right) \tag{55}$$

$$= \log c - \exp\left(\frac{2M^2}{\sigma^2}\right)\left(\mathbb{E}_{p_{pos}(\boldsymbol{z},\boldsymbol{z}')}[k(\boldsymbol{z}, \boldsymbol{z}')] - \mathbb{E}_{p(\boldsymbol{z})p(\boldsymbol{z}')}[k(\boldsymbol{z}, \boldsymbol{z}')]\right). \tag{56}$$

Using the result of lemma 1, we conclude

$$\lim_{n\to\infty} \ell_{sn}(n, 2\sigma^2) \geq \log c - c\exp\left(\frac{2M^2}{\sigma^2}\right) \mathrm{HSIC}(Y, Z). \tag{57}$$

$$\square$$

# F   Appendix for Experiments

## F.1   Details on Experimental Setup

### F.1.1   Performance Gap Experiments

Here we describe the experimental setup for the experiments in section 2.2 and section 2.3, demonstrating the performance gap between E2E training and layer-wise training.

We conducted experiments on CIFAR10 and CIFAR100 (Krizhevsky et al., 2009). Both CIFAR10 and CIFAR100 datasets consist of 50000 training images and 10000 test images, but the number of label types differs between 10 and 100. For backbone models, VGG (Simonyan & Zisserman, 2014) and ResNet (He et al., 2016) are used in the experiments. The VGG network is a combination of convolutional layers, batch-norm layers, and max-pooling layers, allowing for training layer-wisely for each convolutional layer. In contrast, ResNet is characterized by the skip-connections. The smallest unit containing a single skip connection, such as two convolution layers for ResNet18 and three convolution layers for ResNet50, is trained as one module in layer-wise training. The middle layer's representation could have more dimensions than that of the last layer of the backbone network, and it would have computational difficulty to directly flatten the middle representation and calculate similarity on it or pass it to the linear or MLP auxiliary network. To handle this, we applied appropriate average pooling so that the output dimension size would be 2048 before performing the calculation. We trained the model for 400 epochs with the Adam optimizer (Kingma & Ba, 2014); the learning rate was selected from $\tau \in \{0.0005, 0.001\}$, and the learning rate scheduling was performed to decay the learning rate by 0.2 at $[200, 300, 350, 375]$ epochs. For SupCon loss, the model was trained for 1000 epochs with a learning rate $\tau = 0.0005$ due to the long time to convergence, and the learning rate was decayed at $[700, 800, 900]$ by 0.2. Furthermore, the hyperparameter for weight decay was chosen from $\eta \in \{0.0, 0.0005\}$. It prevents overfitting to the training data, and it was especially effective when using auxiliary networks with multiple layers.

### F.1.2  Normalized HSIC Experiments

In the nHSIC experiments in section 4, RBF kernels are employed for input $X$ and intermediate representation $Z$, while a linear kernel is used for label $Y$. The RBF kernel, $k(\boldsymbol{v}, \boldsymbol{w}) = \exp\left(-\|\boldsymbol{v} - \boldsymbol{w}\|_2^2/(2\sigma^2)\right)$, has a $\sigma$ value of $5\sqrt{d}$ following the existing research (Ma et al., 2020; Wang et al., 2021; Jian et al., 2022; Wang et al., 2023b). Additionally, we use an empirical estimator for nHSIC as $\text{Tr}\left[\boldsymbol{KH}(\boldsymbol{KH} + \epsilon m I_m)^{-1}\boldsymbol{LH}(\boldsymbol{LH} + \epsilon m I_m)^{-1}\right]$, where $\epsilon$ is a small constant; we set 1e-5 for this experiment. Although the nHSIC estimator demonstrates consistency (Fukumizu et al., 2007), this study derives the estimator by averaging values based on mini-batches due to computational limitations. Since the estimates calculated from the mini-batch are basically biased estimator, we avoid direct comparison of nHSIC values among layers and instead focus on the dynamics of nHSIC values. Please note that the training dynamics of nHSIC values itself is the comparison of nHSIC values in the same layer.

### F.2  Comparison between E2E and Layer-wise for CIFAR100

Table 4 shows the results for E2E training and layer-wise training with the CIFAR100 dataset. Similar to the case of CIFAR10, the performance of E2E training improves as the model size increases, whereas the performance enhancements of layer-wise training are limited. In the case of directly optimizing the intermediate layers, as in the row of 0-layer, it was challenging to train effectively on the CIFAR100 dataset.

### F.3  Linear Separability for SupCon Loss

Figure 9 compares the linear separability between E2E and layer-wise training; the SupCon loss is adopted for both training methods. When the cross-entropy loss is used as in the main text, the linear separability of E2E training gradually improves, whereas the separability starts to saturate at the relatively early layers. The difference from using the cross-entropy loss is that in layer-wise training, the linear separability at the final layer does not reach 100% for the training data, resulting in a smaller gap between the separability of training and test data. However, the separability for the training data does not improve even if more layers are stacked in the layer-wise training.

### F.4  Additional HSIC Dynamics Experiments

### F.4.1  Other Loss Settings for E2E Training

The purpose of this section is to observe HSIC dynamics using loss functions other than cross-entropy loss used in the paper and to discuss the effect of loss functions. Please note that it does not take into account

Table 4: Test accuracies of E2E training and layer-wise training methods trained on CIFAR100 dataset. The "Auxiliary" column shows the size of auxiliary networks $g$ for layer-wise training. For clarity, the highest accuracy among the different sizes of auxiliary networks is indicated in bold. For CE loss, the middle representations are converted to the space of class number's dimensions; therefore, it does not have 0-layer case.

| | | E2E | | Layer-wise | | | | |
|---|---|---|---|---|---|---|---|---|
| Model | Auxiliary | CE | SupCon | CE | SupCon | Sim | SP (Hard) | SP (Soft) |
| | 0-layer | | | — | 39.0 | 1.7 | 36.5 | 41.7 |
| VGG11 | 1-layer | 68.9 | 64.2 | **60.3** | 55.3 | **51.9** | — | — |
| | 2-layer | | | 57.6 | **62.2** | 50.9 | — | — |
| | 0-layer | | | — | 48.8 | 47.0 | 14.4 | 1.1 |
| ResNet18 | 1-layer | 73.5 | 75.2 | **64.9** | 62.4 | 53.0 | — | — |
| | 2-layer | | | 62.3 | **69.8** | **54.4** | — | — |
| | 0-layer | | | — | 52.5 | 21.0 | 5.2 | 1.2 |
| ResNet50 | 1-layer | 76.3 | 78.1 | **67.5** | 65.6 | 58.3 | — | — |
| | 2-layer | | | 65.8 | **70.2** | **60.6** | — | — |

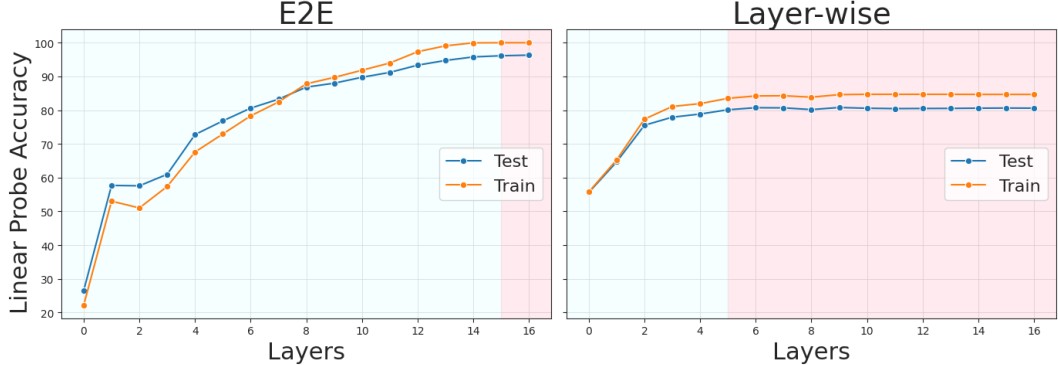

Figure 9: Linear probing accuracies of ResNet50 trained on CIFAR10 dataset. Linear separability for test data and training data are presented. **Left:** E2E training with SupCon loss. **Right:** Layer-wise training with SupCon loss. The model was trained with an auxiliary linear classifier for each block.

the comparison between E2E training and layer-wise training discussed in other sections. As a loss other than cross-entropy, we experimented with the supervised contrastive loss and the self-supervised loss used in SimCLR (Chen et al., 2020), which maximizes the InfoNCE lower bound of mutual information among different views. Figure 10 shows that both of these losses lead to a uniform reduction in nHSIC$(X; Z)$ and nHSIC$(Y; Z)$ without information bottleneck behavior. The supervised contrastive loss obtained the representations with higher information content with the label $Y$ because of the supervision. In contrast, SimCLR-style loss deals with the mutual information among the representations of different views, there is no interpretation provided for the mutual information with labels or inputs as in cross-entropy.

### F.4.2 E2E training with Forward Gradient

We explored the forward gradient method (Baydin et al., 2022) as a backpropagation-free method, other than layer-wise training. Forward gradient updates weights without using backpropagation but optimizes a single loss function for the entire model; therefore, it is E2E training, unlike layer-wise training. Figure 11 compares backpropagation and forward gradient for the LeNet5 model, like figure 3. The forward gradient method showed slow convergence, and the test accuracy for CIFAR10 did not improve sufficiently. The improvement in nHSIC values at each layer is also gradual, particularly the information amount with input

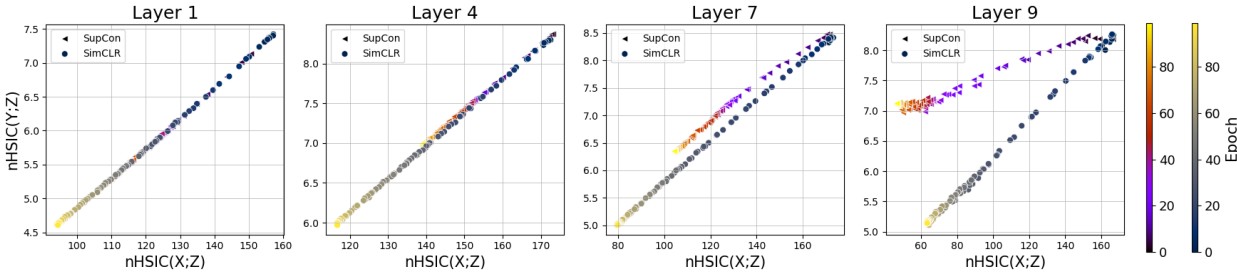

Figure 10: HSIC plane dynamics of E2E training with SimCLR-style loss and supervised contrastive loss. ResNet18 model was trained on CIFAR10 dataset. The color gradation shows the progress of training. Circles with a navy-yellow-based colormap denote SimCLR-style loss, whereas left triangles with a purple-yellow-based colormap show supervised contrastive loss.

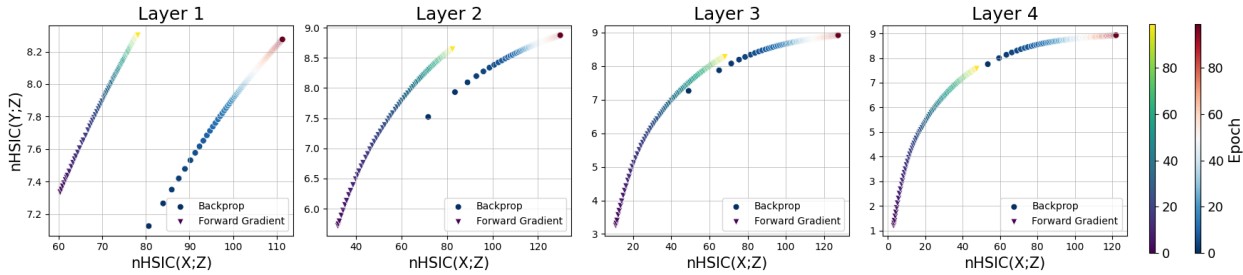

(a) LeNet5 models trained on MNIST dataset. Test accuracy at 100 epoch: 99.0% (E2E with backpropagation), 82.4% (E2E with forward gradient).

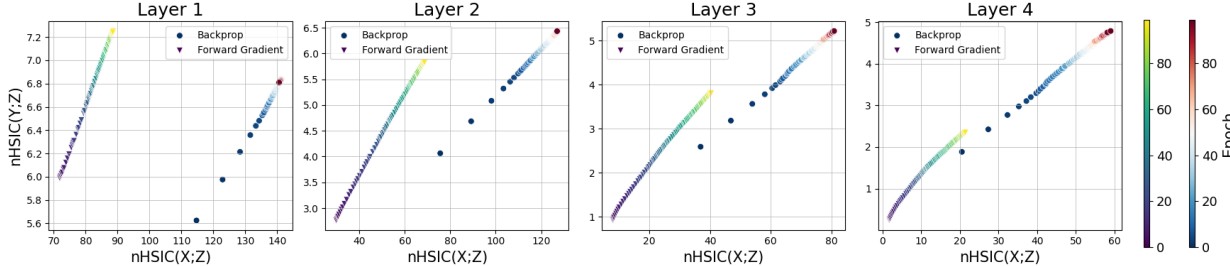

(b) LeNet5 models trained on CIFAR10 dataset. Test accuracy at 100 epoch: 62.2% (E2E with backpropagation), 26.8% (E2E with forward gradient).

Figure 11: HSIC plane dynamics of E2E training with backpropagation and forward gradient method for LeNet5 model. The color gradation shows the progress of training. Inverted triangles with a blue-yellow-based colormap denote forward gradient, whereas circles with a red-based colormap show backpropagation.

$X$ improved slowly. The important point here is that in the same setting of LeNet5 + CIFAR10 as in figure 3, we did not observe early layer information degradation as seen in the layer-wise training model. While the forward gradient method is a backpropagation-free technique but an E2E training method, unlike layer-wise training, the interaction among the layers is possible through forward automatic differentiation.

### F.4.3 Sequential Layer-wise Training

In this section, we experimented with the sequential layer-wise training mentioned in section B.1. The sequential layer-wise training exhibits lower test accuracy than the simultaneous setting as mentioned in the previous section. When ResNet18 is trained on the CIFAR10 dataset using CE loss, table 1 shows that test accuracies of the simultaneous setting are 89.3 and 89.9 for linear and MLP auxiliary classifiers, respectively. In contrast, these test accuracies dropped to 86.6 and 86.2 in the case of sequential training. Here each layer

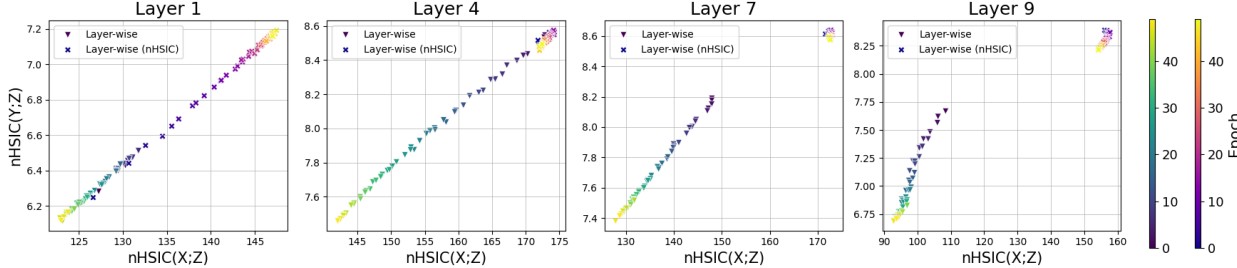

(a) Results for sequential layer-wise training and E2E training. Inverted triangles with a blue-yellow-based colormap denote sequential layer-wise training, whereas circles with a red-based colormap show E2E training.

(b) Results for sequential layer-wise training with HSIC augmenting term. Inverted triangles with a blue-yellow-based colormap denote sequential layer-wise training, whereas cross marks with a red-yellow-based colormap show training with HSIC augmenting term ($\lambda = 0.005$). Test accuracy at 50 epoch: 85.6% (original), 85.1% (nHSIC).

Figure 12: HSIC plane dynamics of sequential layer-wise training. ResNet18 is trained on the CIFAR10 dataset for 50 epochs per layer, resulting in 500 epochs to train the entire model because ResNet18 consists of 10 layer-wise trainable modules. The color gradation shows the progress of training.

is trained for 100 epochs, meaning that the entire model is trained for 1000 epochs given that ResNet18 has 10 layer-wise trainable modules.

On the other hand, it is the same between sequential and simultaneous layer-wise training that nHSIC behavior is uniform among the layers, and there is no information-bottleneck behavior in the final layer (see figure 12). In the case of nHSIC regularization, the improvement in information content between $X$ and $Z$, as well as $Y$ and $Z$, because of the nHSIC augmenting term was more significant compared to simultaneous training in section 4.4. This can be attributed to the fact that the sequential layer-wise training exactly corresponds to the greedy layer-wise optimization, which was discussed in section 4.2, and the nHSIC regularization can prevent the severe corruption of the input information. However, there was no improvement in accuracy, which was the same as the simultaneous layer-wise training.

### F.4.4   Model Settings

**Toy-MLP.**   We discussed the CNN model LeNet5 in section 4.1, while here, we experimented with MLP. Figure 13 shows the HSIC dynamics for five-layer MLP with the hidden size of 512. In the case of MNIST, there is no difference in test accuracy between E2E training and layer-wise training, similar to LeNet5. However, E2E training learns the inputs and labels slowly, which could be attributed to the distance of the backpropagation path from the loss evaluation. In the case of CIFAR10, layer-wise training exhibited slower improvement in nHSIC values as layers progressed like the LeNet5 case; however, there is no information compression in the initial layer observed in LeNet5. In MLP models, please note that training the first layer in a layer-wise manner with a linear auxiliary classifier is equivalent to E2E training of a two-layer MLP model.

**Vision Transformer.**   Figure 14 shows the HSIC dynamics for vision transformer (ViT) (Dosovitskiy et al., 2020) with six transformer blocks, the patch size of 4, and the hidden dimension of 512. The local

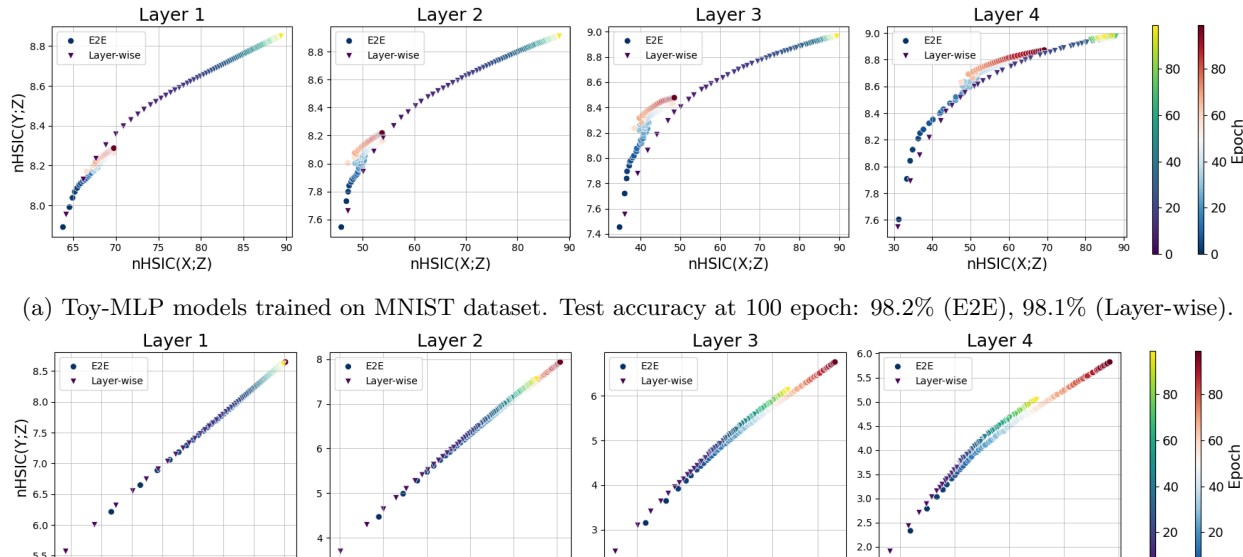

(a) Toy-MLP models trained on MNIST dataset. Test accuracy at 100 epoch: 98.2% (E2E), 98.1% (Layer-wise).

(b) Toy-MLP models trained on CIFAR10 dataset. Test accuracy at 100 epoch: 58.7% (E2E), 57.8% (Layer-wise).

Figure 13: HSIC plane dynamics of Toy-MLP model. The color gradation shows the progress of training, i.e., the number of epochs. Inverted triangles with a blue-yellow-based colormap denote layer-wise training, whereas circles with a red-based colormap show E2E training.

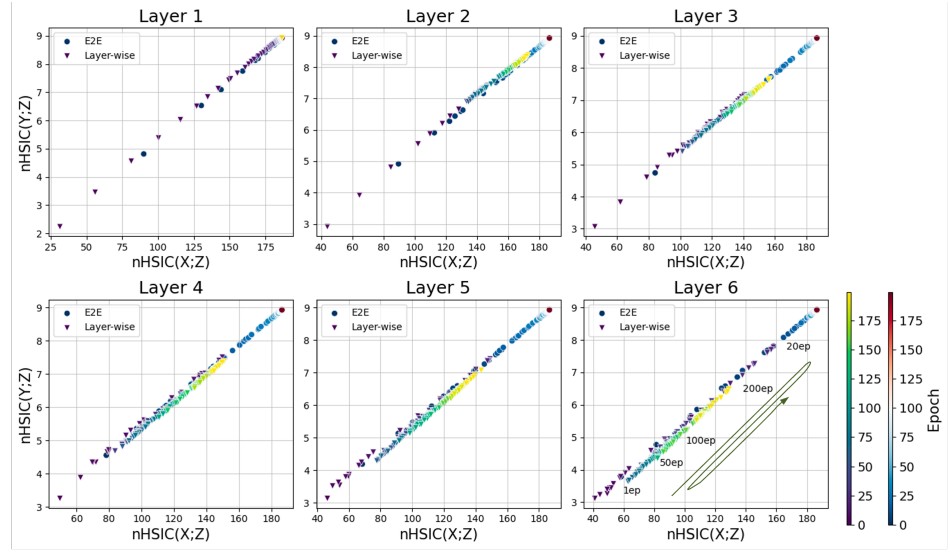

Figure 14: HSIC plane dynamics of ViT model trained on CIFAR10 dataset. The color gradation shows the progress of training. For complex behavior of layer-wise training, the number of epochs and dynamics are annotated in the output layer. Inverted triangles with a blue-yellow-based colormap denote layer-wise training, whereas circles with a red-based colormap show E2E training.

block-wise training results in information compression and leads to worse test accuracy, as with the results presented in the main text. Conversely, E2E training shows no information compression in both middle and output representations, meaning no IB behavior. The test accuracy of trained ViT is not sufficiently high for our problem setting, and further validation in larger-scale settings will be necessary to conclude.

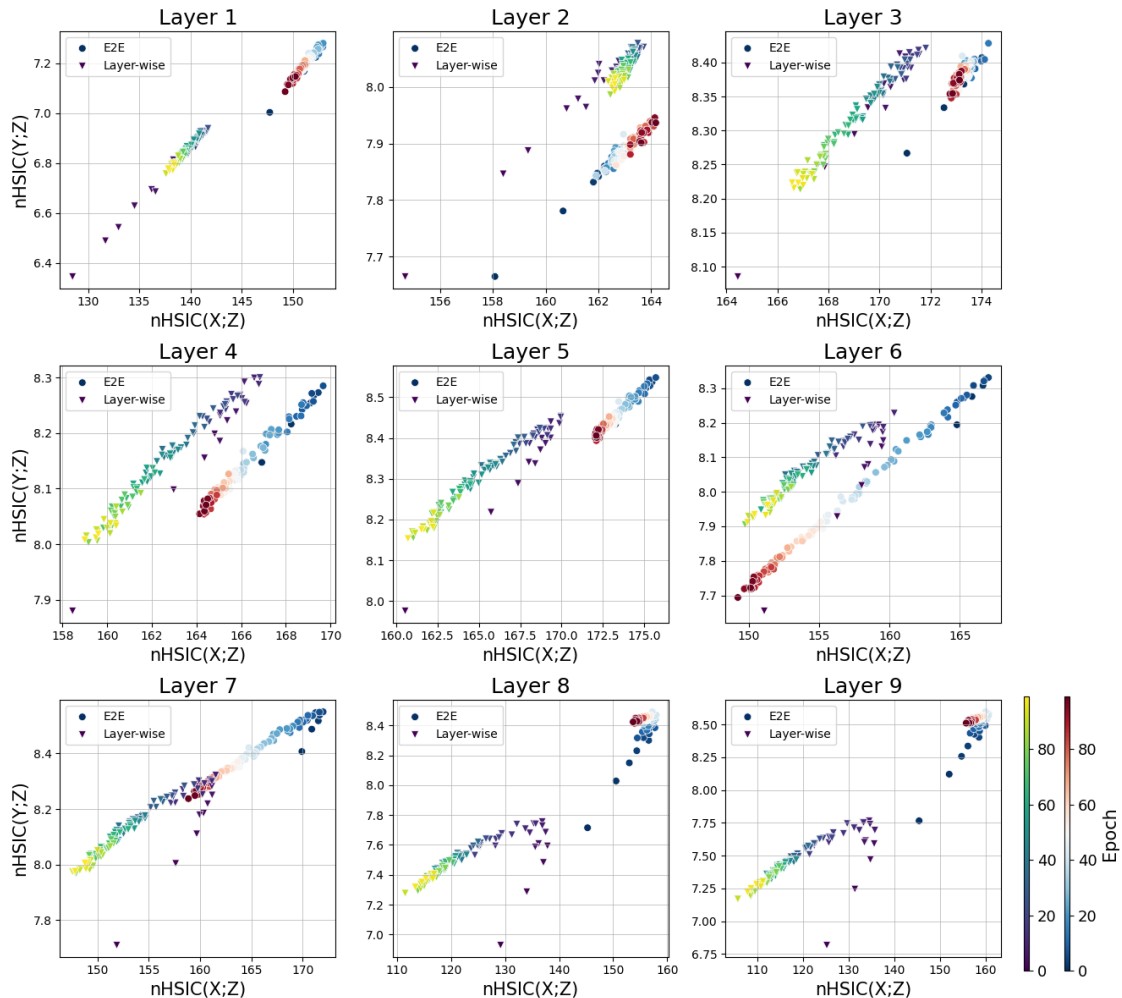

Figure 15: HSIC plane dynamics of ResNet18 model trained on CIFAR10 dataset. The color gradation shows the progress of training. Inverted triangles with a blue-yellow-based colormap denote layer-wise training, whereas circles with a red-based colormap show E2E training.

**ResNet18.** The HSIC dynamics for all layers of ResNet18 are illustrated in figure 15. As observed in figure 6, while layer-wise training showed compression in all layers, the E2E training compressed the intermediate layers while maintaining nHSIC$(Y, Z)$ at the layer in the output side. This HSIC preservation was seen in the last two layers, which corresponds to the last module group when classifying components of ResNet18 by the size of output channels.

**ResNet50.** The HSIC dynamics for all layers of ResNet50 are presented in figure 16. ResNet50 has 17 intermediate representations, and the layer-wise training demonstrates compression across all layers. In the E2E training, the HSIC reduction in the middle layers is less observed than in the ResNet18 setting, and it primarily occurs in the early layers. Additionally, the final block not only preserves nHSIC$(Y, Z)$ but also increases it while reducing nHSIC$(X, Z)$, showing a behavior closer to the IB hypothesis.

**VGG11.** The results for VGG11 are shown in the figure 17. In this setting, the layer-wise training precisely trains a single convolutional layer, whereas a block with one skip-connection in the ResNet setting. The overall behaviors are similar to those of ResNet. The E2E training demonstrates the compression in the intermediate layers while exhibiting behavior suggested by IB hypothesis in the output layers. The layer-wise

training shows little compression of HSIC values in the output layers, differing from ResNet, but generally presents the same low nHSIC values as in the ResNet case.

**VGG11 without batch normalization.**   We made an intriguing observation regarding the compression in the middle layers for E2E training. In figure 18, while layer-wise training includes batch normalization layers and is the same setting as before, for the E2E trained model, we excluded the batch normalization layers. This configuration without batch normalization is often used in the VGG network. Interestingly, we observed no compression in the nHSIC value for the intermediate and output layers. The final test accuracy was higher than that of layer-wise training but lower than when batch normalization was included. This suggests the potential role of batch normalization in compressing intermediate representations. There is a possibility that the E2E training could cooperatively optimize the scales of batch normalization layers across layers. Further exploration is a future work.

**ResNet18 with HSIC Augmenting Term.**   The figure 19 shows the results of layer-wise training with an additional term aimed at increasing nHSIC values. While this modification can prevent compression in HSIC dynamics, the obtained test accuracy deteriorated. The smaller $\lambda$ values result in the nHSIC behavior more similar to the original layer-wise training. Please note that we used a batch size of 256 here instead of the batch size of 128 in the original training due to the effect of nHSIC estimation. As discussed in the main text, the behavior of HSIC dynamics fundamentally remains similar to that of layer-wise training, revealing a gap from the E2E training in terms of both the HSIC dynamics and the final performance.

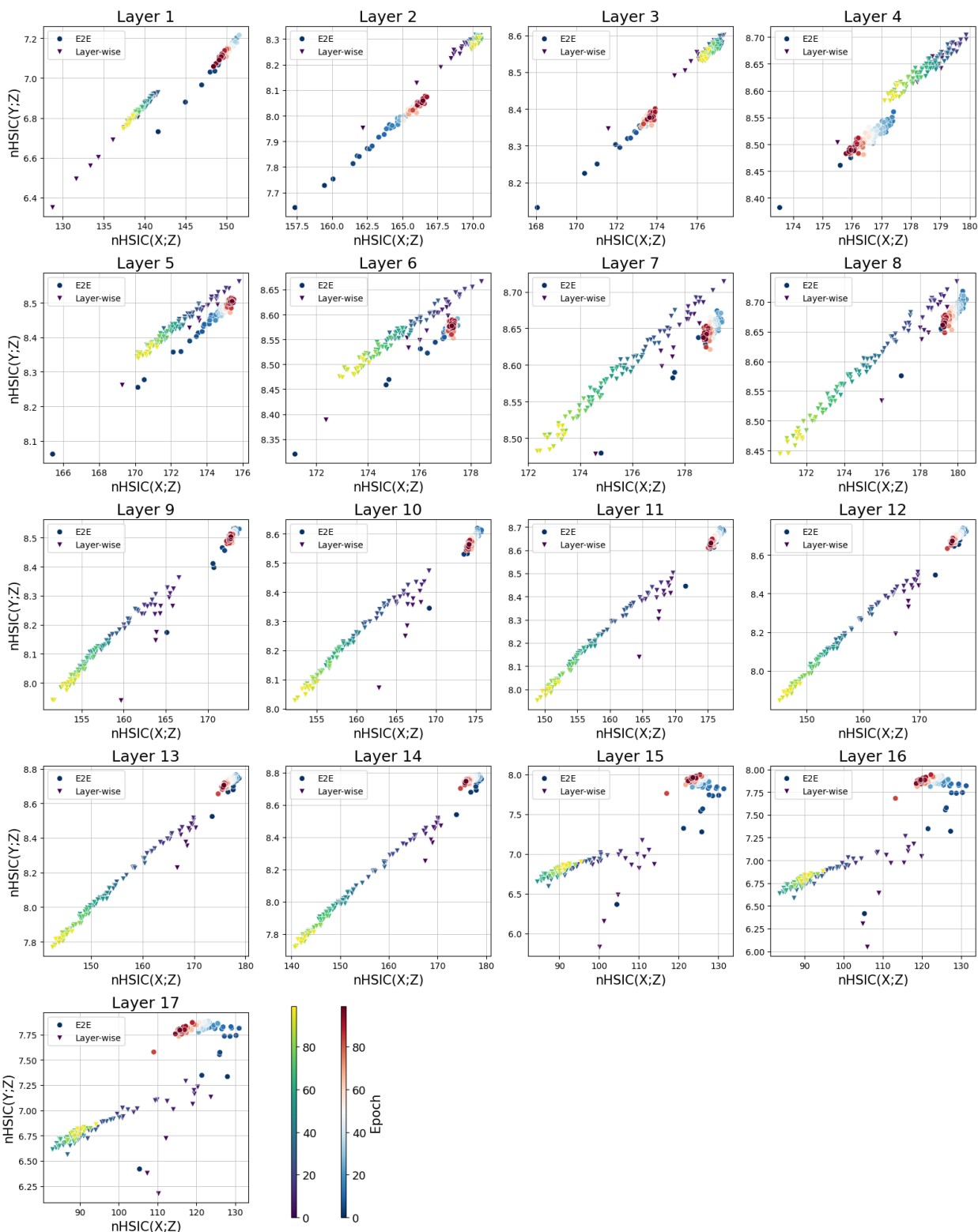

Figure 16: HSIC plane dynamics of ResNet50 model trained on CIFAR10 dataset. The color gradation shows the progress of training. Inverted triangles with a blue-yellow-based colormap denote layer-wise training, whereas circles with a red-based colormap show E2E training.

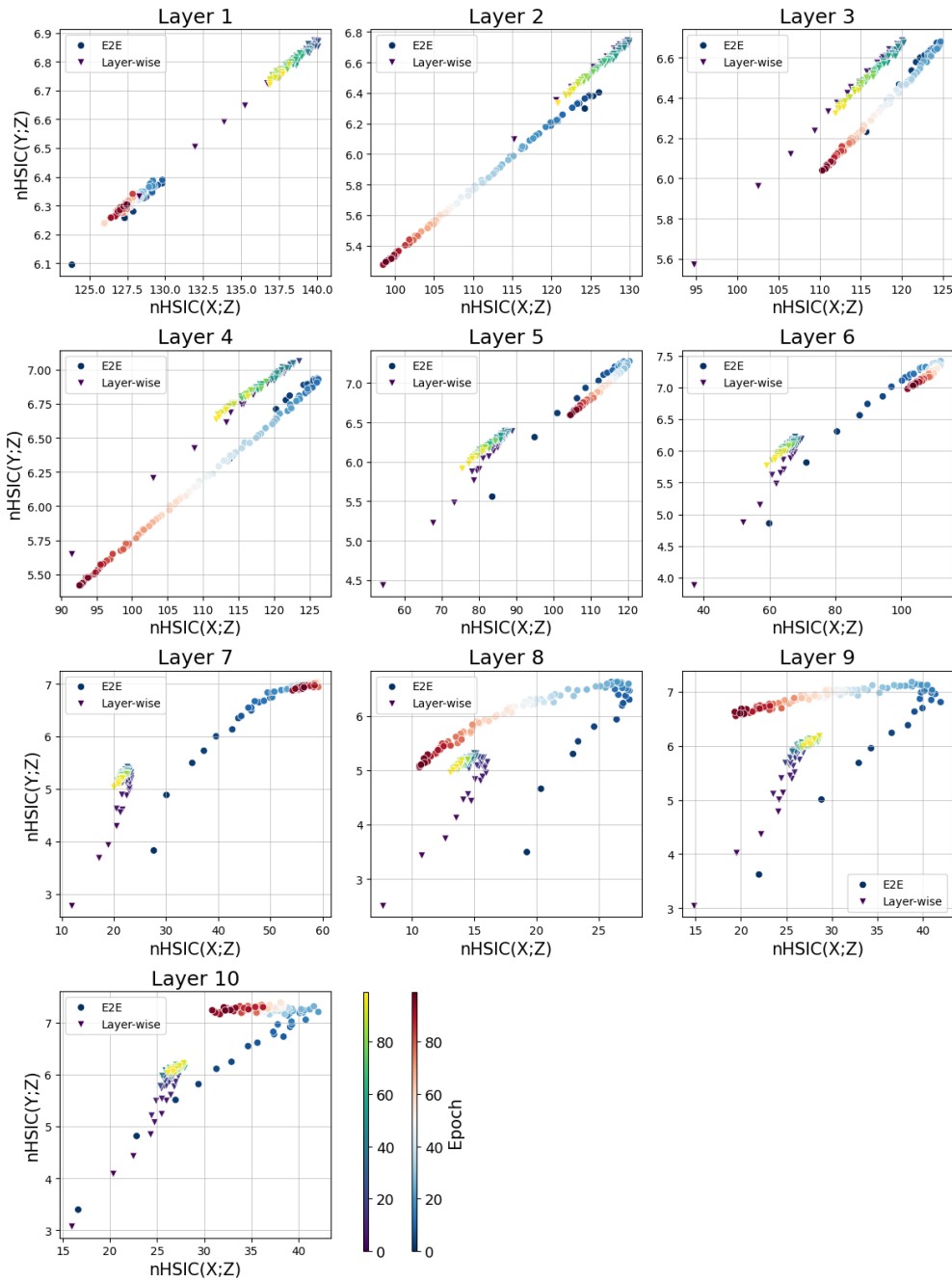

Figure 17: HSIC plane dynamics of VGG11 model trained on CIFAR10 dataset. The color gradation shows the progress of training. Inverted triangles with a blue-yellow-based colormap denote layer-wise training, whereas circles with a red-based colormap show E2E training.

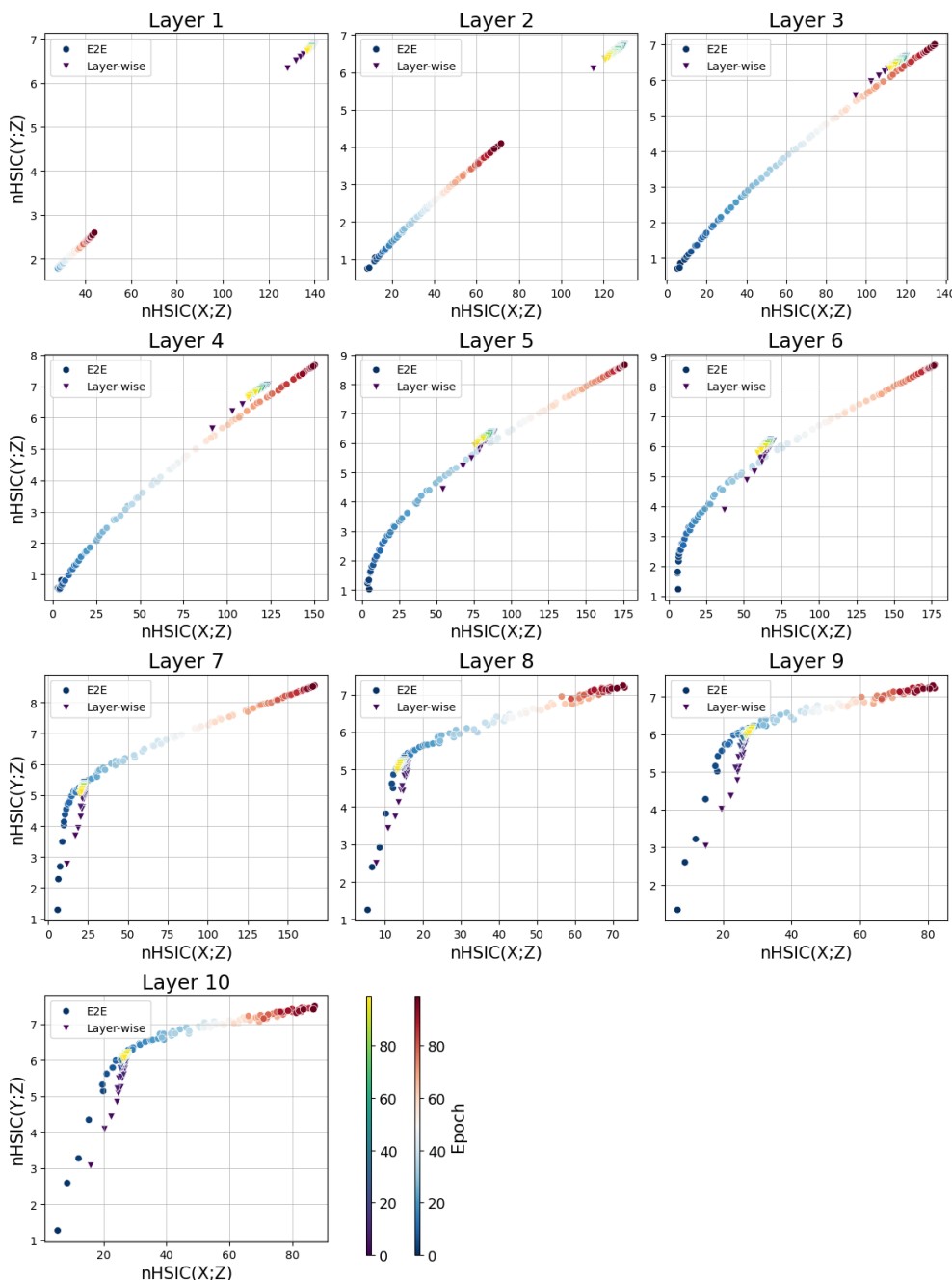

Figure 18: HSIC plane dynamics of VGG11 model trained on CIFAR10 dataset. In this figure, the model trained in the E2E manner does not contain the batch normalization layers. The color gradation shows the progress of training. Inverted triangles with a blue-yellow-based colormap denote layer-wise training, whereas circles with a red-based colormap show E2E training.

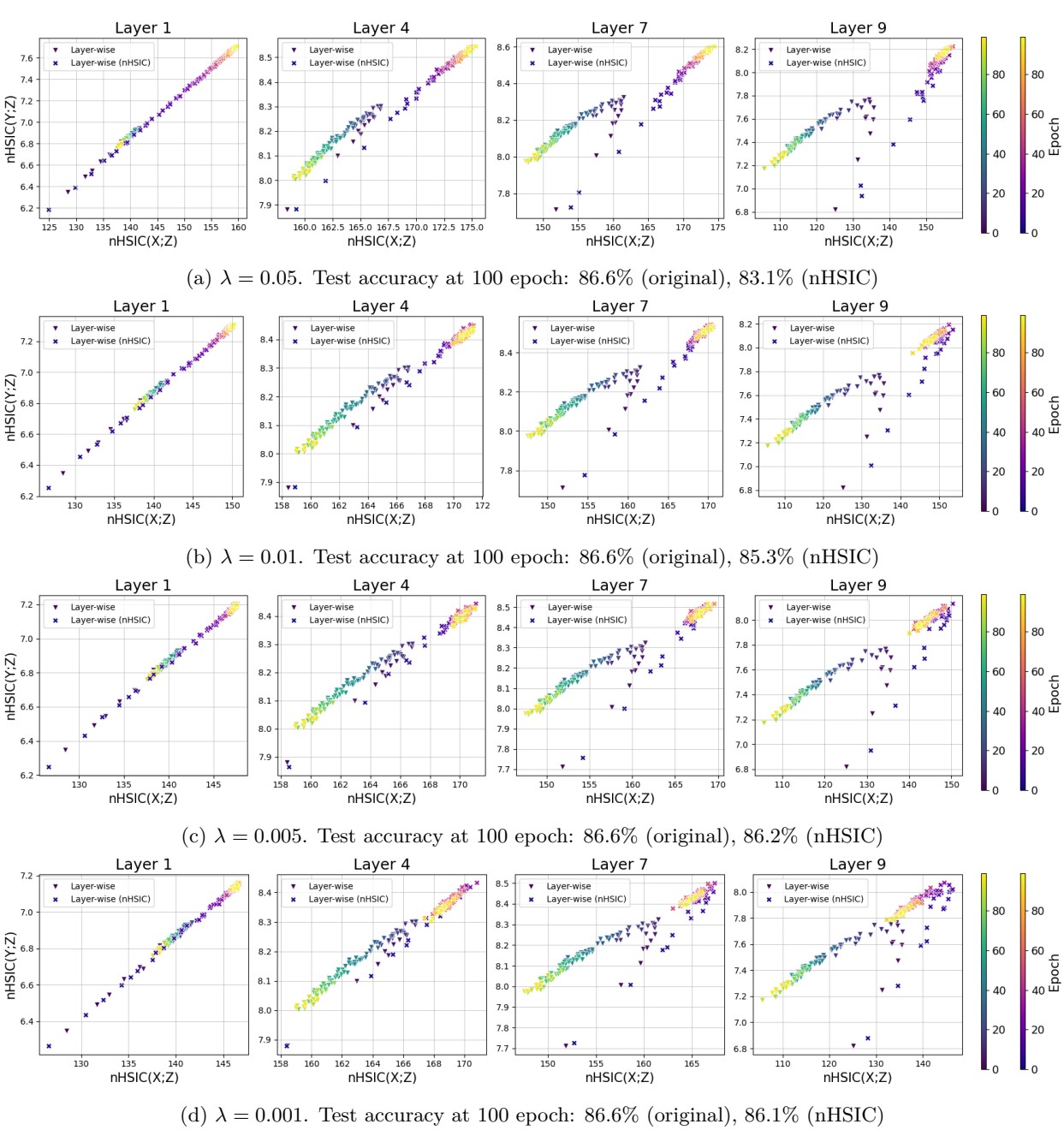

(a) $\lambda = 0.05$. Test accuracy at 100 epoch: 86.6% (original), 83.1% (nHSIC)

(b) $\lambda = 0.01$. Test accuracy at 100 epoch: 86.6% (original), 85.3% (nHSIC)

(c) $\lambda = 0.005$. Test accuracy at 100 epoch: 86.6% (original), 86.2% (nHSIC)

(d) $\lambda = 0.001$. Test accuracy at 100 epoch: 86.6% (original), 86.1% (nHSIC)

Figure 19: HSIC plane dynamics for layer-wise training with HSIC augmenting term for different $\lambda$ values. ResNet18 is trained on CIFAR10. The results of four layers are summarized in this figure. The color gradation shows the progress of training. Inverted triangles with a blue-yellow-based colormap denote layer-wise training in the original setting as a baseline, whereas the cross marks with a red-yellow-based colormap show training with HSIC augmenting term.

