# OpenReview forum: "End-to-End Training Induces Information Bottleneck through Layer-Role Differentiation: A Comparative Analysis with Layer-wise Training"
_TMLR — Accepted by TMLR_

### Review · Reviewer_cn4C · 2024-03-02

**Summary Of Contributions:**

The paper explores the efficacy of end-to-end (E2E) training versus layer-wise training in the learning of deep neural networks. It looks into how E2E training excels in information propagation and assigns diverse roles to layers, leading to an efficient model representation. Through information-theoretic analysis and HSIC metrics, the study demonstrates that E2E training promotes better information flow and layer-role differentiation. It suggests that these advantages contribute to the observed performance benefits of E2E training, offering a new perspective and insight on neural network training dynamics.

**Audience:**

Yes

**Claims And Evidence:**

Yes

**Requested Changes:**

Please see the weakness part. Specifically, including the analysis with the transformer architecture could improve the scope of the paper.

**Strengths And Weaknesses:**

### **Strength**
- The paper is overall organized, facilitating a smooth reading experience. Additionally, the illustrative figures and experimental visualizations simplifies the understanding of the analysis.

- The mathematical analysis are technically sound. It effectively demonstrates how E2E training promotes better information flow and differentiation of layer roles, leading to superior model performance.

- This comparative analysis offers a fresh perspective on neural network training dynamics, contributing valuable insights to the field of deep learning research.

### **Weakness**
- Although the paper leverages the analysis in the lens of information theory, specifically of mutual information, it mostly discuss the observation in terms of information compression with model trained with cross-entropy. How would the model behave when using mutual-information-oriented loss, e.g. the one proposed in Hjelm, R. Devon, et al. 2018?

- The paper only demonstrates the results on convolutional model architectures for its conclusions, this could limit the universality of its findings. It could be beneficial to valid the analysis on recent transformer architecture, pure MLP architecture, etc.

- Considering the models studied in the paper are limited, and some recently popular architectures such as ViT, Swin transformer, ConvNext are not studied. The paper might not fully explore the practical implications of its findings, particularly how they could be applied in real-world scenarios or influence the development of future deep learning models.


### References

- Hjelm, R. D., Fedorov, A., Lavoie-Marchildon, S., Grewal, K., Bachman, P., Trischler, A., & Bengio, Y. (2018). Learning deep representations by mutual information estimation and maximization. arXiv preprint arXiv:1808.06670.

---

> ### Author Response · Authors · 2024-03-11
> **Response to Reviewer cn4C**
>
> Thank you for your constructive reviews! We appreciate your consideration to improve the work.
> >  (Weakness 1) Although the paper leverages the analysis in the lens of information theory, specifically of mutual information, it mostly discuss the observation in terms of information compression with model trained with cross-entropy. How would the model behave when using mutual-information-oriented loss, e.g. the one proposed in Hjelm, R. Devon, et al. 2018 ?
>
> We have newly included results for the information-oriented loss other than cross-entropy.  \
> In this study, the use of cross-entropy loss is motivated by the fact the optimization of cross-entropy leads to the maximize the variational lower-bound of $I(Z; Y)$, as discussed in section E.1. We state that despite maximizing this lower-bound, the E2E training of relatively deep network implicitly leads to information reduction between input $X$ and $Z$.
>
> As a loss other than cross-entropy, we have added experiments with the self-supervised loss used in SimCLR [1], which maximizes the InfoNCE lower-bound of mutual information among different views. It shares the mind with DeepInfomax (DIM) [2], as DIM aims to extract features by maximizing MI between the global and local features, while SimCLR loss can be adopted with the uncustomized ResNet and does not require complex hyper-parameter tuning [3]. The results of SimCLR loss for E2E training show a uniform reduction in nHSIC(X; Z) and nHSIC(Y; Z) without information bottleneck behavior. It is important to note that since this self-supervised loss deals with the mutual information between representations of different views, there is no interpretation provided for the mutual information with labels or inputs as in cross-entropy.
>
> [1]  Chen+, "A simple framework for contrastive learning of visual representations.", 2020. \
> [2] Hjelm+, "Learning deep representations by mutual information estimation and maximization.", 2018. \
> [3] Falcon+, "A framework for contrastive self-supervised learning and designing a new approach", 2020.
>
>
> >  (Weakness 2) The paper only demonstrates the results on convolutional model architectures for its conclusions, this could limit the universality of its findings. It could be beneficial to valid the analysis on recent transformer architecture, pure MLP architecture, etc.
>
> We have added the results for the pure MLP and transformer architecture to the appendix. \
> In the  MLP model, conducting layer-wise training in the first layer with an auxiliary linear network is equivalent to training an MLP with a single hidden layer in an E2E manner. While no significant differences were observed between E2E training and layer-wise training, similar to the results in the main text, layer-wise training exhibited limited improvement in nHSIC values as layers progressed.
> For the details of transformer results, please refer to the next.
>
>
> >  (Weakness 3) Considering the models studied in the paper are limited, and some recently popular architectures such as ViT, Swin transformer, ConvNext are not studied. The paper might not fully explore the practical implications of its findings, particularly how they could be applied in real-world scenarios or influence the development of future deep learning models.
>
> We have added nHSIC dynamics for ViT with six transformer blocks to the appendix. \
> As with the results presented in the main text, local block-wise training results in information compression and leads to worse test accuracy. Conversely, E2E training shows no information compression in both middle and output representations, meaning no IB behavior.
> In section F.2.3, we mentioned that the existence of batch normalization affects IB behavior, and it is possible that differences in the transformer architecture lead to this no IB behavior. However, the performance of trained ViT is not sufficiently high for our problem setting [4], and it seems that further validation in larger-scale settings will be necessary to conclude.
>
> This paper explores the benefits of E2E training through comparison with layer-wise training from an information-theoretic perspective. Rather than the development of new deep learning models, we believe it contributes to a better understanding of the behavior and effectiveness of deep learning models and E2E training.
>
> [4] Zhu+, "Understanding Why ViT Trains Badly on Small Datasets: An Intuitive Perspective", 2023.

---

### Review · Reviewer_CGUu · 2024-04-09

**Summary Of Contributions:**

The paper proposes an analysis of end-to-end training of neural networks as compared to layer wise training. In particular, the authors consider the information flow in the networks and analyse it through the Hilbert-Schmidt independence criterion (HSIC) dynamics in different layers in networks trained either end-to-end or layer-wise.

Beyond confirming the performance gap in different models, this analysis revealed 2 main observations: With end-to-end training, middle representations are more entangled but exhibit higher correlation with the labels. Moreover, with end-to-end training, the HSIC dynamics are different across layers, and the layers play different roles in information compression, while layer-wise training shows a more uniform behavior.

The authors additionally proposed to use HSIC to regularise layer-wise training to increase information preservation across layers, but this led to worse test performance. The advantages of end-to-end training are further discussed from an information bottleneck and from a geometric perspective.

**Audience:**

Yes

**Broader Impact Concerns:**

The authors discuss the limitations of their work. Given the theoretical nature of the work, this discussion is interesting and sufficient.

**Claims And Evidence:**

Yes

**Requested Changes:**

From the first point under weaknesses:
* Can the authors clarify the layer-wise training procedure? Is it done jointly for all layers using the objective (2), or is it done sequentially as in e.g. "Greedy Layerwise Learning Can Scale to ImageNet" Belilovsky et al. 2019? If it's the former, is it evaluated using only the last layer output or by pooling outputs from intermediary layers too? Either way, it would be interesting to compare these two alternatives and their impact on the performance gap, and with the HSIC regularization.
* Can the authors analyse other backpropagation free baselines?

From the second point under weaknesses:
* Can the authors show how their observations vary when varying data and models?

**Strengths And Weaknesses:**

Strengths:
* The paper reveals an interesting analysis of the information flow in the networks, and leads to insights that are novel to the best of my knowledge.
* The use of HSIC is judicious, and provides a criterion that can be useful for further studies in developing backpropagation free approaches.
* The paper is well structured, and clearly written, and the discussion is of relevance to the community.

Weaknesses:
* The end-to-end alternative studied here is really simple and lacks some clarity (more details below). It would be interesting to consider other backpropagation alternatives and study their impact. An example that comes to mind is the "forward gradient" formulation (see e.g. Gradients without Backpropagation, Baydin et al. 2022).
* It seems that more empirical validation is in order. For example, concerning the analysis in Figure 1, it would be interesting to study the impacts of varying data (complexity of the input, number of classes,...) and models (containing convolutions, skip connections, attention mechnisms, etc...).

---

> ### Author Response · Authors · 2024-04-20
> **Response to Reviewer CGUu**
>
> Thank you for the review! We appreciate your thorough consideration of improving the work.
> > (1) Can the authors clarify the layer-wise training procedure? Is it done jointly for all layers using the objective (2), or is it done sequentially as in e.g. "Greedy Layerwise Learning Can Scale to ImageNet" Belilovsky et al. 2019? If it's the former, is it evaluated using only the last layer output or by pooling outputs from intermediary layers too? Either way, it would be interesting to compare these two alternatives and their impact on the performance gap, and with the HSIC regularization.
>
> We consider the setting that each layer is trained simultaneously, and the model's prediction is made by the last output.
> This is because the primary focus of our research is on comparing with E2E training, thus we filled the gap between E2E training and layer-wise training except for the granularity of loss functions.
> To clarify this point, we have added the sentence **"In layer-wise training, each layer is trained simultaneously, and the model's prediction is made by using only the last output (see section B.1 for training details)."** to the first block of section 2.1, and we have also included a more detailed explanation in appendix B.1.
>
> Additionally, we newly experimented with this sequential layer-wise training in toy-CNN and resnet18 settings.
> Compared with the simultaneous layer-wise training in the main text, the model performance deteriorated as stated in [1, 2, 3], however, it is the same that nHSIC behavior is uniform among the layers and there is NO information-bottleneck behavior in the final layer.
> In the case of nHSIC regularization, the improvement in information content between $Z$ and $X$, as well as $Z$ and $Y$, because of the nHSIC augmenting term was more significant compared to simultaneous training. This can be attributed to the fact that the sequential layer-wise training exactly corresponds to the greedy layer-wise optimization, which was discussed in section 4.2, and the nHSIC regularization can prevent the severe corruption of the input information. However, there was no improvement in accuracy, which was the same as the simultaneous layer-wise training.
> We have already included the results in the appendix.
>
> [1] Löwe, "Putting An End to End-to-End: Gradient-Isolated Learning of Representations", 2019. \
> [2] Wang+, "Revisiting locally supervised learning: an alternative to end-to-end training", 2020. \
> [3] Siddiqui+, "Blockwise self-supervised learning at scale", 2023.
>
> > (2) Can the authors analyse other backpropagation free baselines?
>
> We have newly included results for the forward gradient method in the appendix as another backpropagation free baseline.
> We experimented with MNIST dataset following the original paper settings [4]. The information planes of nHSIC values were newly shown for toy four-layers MLP and LeNet5, like figure3 in our paper.
>
> The forward gradient method showed slow convergence, particularly noticeable in the case of LeNet5 where there was a significant difference in test accuracy within the same epoch.  In the analysis of the information plane, we doubled the number of epochs for the forward gradient method, and it was confirmed that the improvement in information content at each layer between $Z$ and $X$, as well as $Z$ and $Y$, is slower compared to E2E training. The important point here is that in the same setting of LeNet5 + CIFAR10 as in figure 3, we did not observe early layer information degradation as seen in the layer-wise training model. While the forward gradient method is a backpropagation-free technique, unlike layer-wise training, the interaction among the layers is possible through forward automatic differentiation. We suppose it strengthens our claim in this paper.
>
> [4] Baydin+, "Gradient without Backpropagation", 2022.
>
> > (3) Can the authors show how their observations vary when varying data and models?
>
> The comparison between E2E training and layer-wise training for CIFAR100 dataset has been shown in appendix F.2. The superiority of E2E training is still consistent when the number of classes is increased.
> Furthermore, we have newly included the information plane results for the pure MLP and ViT models besides the CNN models given the discussion with other reviewers. We believe that it further broadens the range of our work.

---

### Review · Reviewer_tneY · 2024-04-14

**Summary Of Contributions:**

The paper analyzed the difference between end-to-end training and layerwise training from the information bottleneck perspective. It mainly compared normalized HSIC - nHSIC(X,Z) and nHSIC(Y,Z) of representations z at different layers. The paper found end-to-end training is able to maintain high HSIC in the final layer.

**Audience:**

Yes

**Broader Impact Concerns:**

No broader concerns.

**Claims And Evidence:**

No

**Requested Changes:**

Please address the questions in the weakness.

**Strengths And Weaknesses:**

Strength:

The idea of explaining the representations of different neural network layers with HSIC is insightful.

Weakness:

-  The writing is unclear. It is hard to understand the argument of the paper. What does the paper prove? Why they are important? For example, the introduction discussed " the human brain has connections on the order of 100 trillion, whereas GPT-4 has roughly 1 trillion parameters". Why is this relevant to this paper?

- Comparing E2E training with layer-wise training is not well-motivated. How does it help to understand the information bottleneck by E2E training? How does the local optimization phenomenon observed in layer-wise training support the IB theory of E2E training?

- Instead of showing the HSIC of different epochs, it will be more straightforward to visualize those of different layers in the same plot. In the paper the x and y scales of different layers are not shared.

---

> ### Author Response · Authors · 2024-04-20
> **Response to Reviewer tneY (1/1)**
>
> Thank you for the constructive questions. We hope this rebuttal has answered all your questions.
>
> > (Weakness 1) The writing is unclear. It is hard to understand the argument of the paper. What does the paper prove? Why they are important? For example, the introduction discussed " the human brain has connections on the order of 100 trillion, whereas GPT-4 has roughly 1 trillion parameters". Why is this relevant to this paper?
>
> > 1.1. What does the paper prove?
>
> This paper empirically proves that the advantages of E2E training are 1) input information propagation and 2) layer-role differentiation, leading to information bottleneck (IB) representation in the final layer.
> We also prove that the curious compression of HSIC(Z; Y), which was observed in the middle layers with E2E training, leads to the increase of soft nearest neighbor loss [1] and higher model performance according to their work [1].
>
> > 1.2. Why they are important?
>
> The above finding is important because it newly provides the advantages of E2E training from the perspectives of input information propagation and IB. It also provides the limitations of layer-wise training and the criteria that backpropagation-free training methods should meet.
> Furthermore, from the perspective of understanding the principles of IB in deep learning, they are important as they suggest the need to consider the cooperative interactions among layers, not just the final layer.
>
> To clarify the contributions of this paper and why they are important, we have updated the following points in the main text.
> - (In abstract, update) "**Our work not only provides the advantages of E2E training in terms of information propagation and the information bottleneck but also** suggests the need to consider the cooperative interactions between layers, not just the final layer when analyzing the information bottleneck of deep learning."
> - (In section 1, update) "Additionally, our HSIC analysis shows the information bottleneck behavior in the final layer only for E2E training. **It shows the benefit of E2E training in terms of information bottleneck and suggests a connection between the middle layer compression in E2E training brought by layer-role differentiation and the acquisition of information bottleneck representation at the final layer.**"
> - (In conclusion, update) "assigns diverse roles to the layers, **leading to the information bottleneck representation of the final layer.**"
> - (In conclusion, update) "The **information compression of the intermediate layers** observed in this study"
> - (In conclusion, update) "the information compression mechanisms **of the middle layers**"
>
> > 1.3. Why is this relevant to this paper?
>
> This is relevant because It introduces the motivation to focus on the advantages of E2E training, which is the primary purpose of our work.
> This part comparing the number of human brain's connections to the number of LLM parameters refers to the parameter efficiency and high performance of recent LLMs compared to the human brain [2].
> It gives the possibility that biologically plausible methods are not necessarily better than E2E training with backpropagation, and It motivates to focus on the advantages of E2E training itself rather than proposing backpropagation alternatives that are not trained in E2E manner (see Appendix A for the recent work on backpropagation alternatives).
>
> [1] Frosst, "Analyzing and improving representations with the soft nearest neighbor loss", 2019. \
> [2] Hinton, “Godfather of AI”, quits Google to warn of AI risks (Host: Pieter Abbeel)", 2023.

---

> ### Author Response · Authors · 2024-04-20
> **Response to Reviewer tneY (2/2)**
>
> >  (Weakness 2) Comparing E2E training with layer-wise training is not well-motivated. How does it help to understand the information bottleneck by E2E training? How does the local optimization phenomenon observed in layer-wise training support the IB theory of E2E training?
>
> > 2.1. Comparing E2E training with layer-wise training is not well-motivated. How does it help to understand the information bottleneck by E2E training?
>
> Layer-wise training shares fundamental learning principles and architectures with E2E training, with the granularity of loss evaluation being the only difference. Therefore, we can reveal the benefits of being E2E by comparing E2E training with layer-wise training, which is the same training method except for training in an E2E manner.
> In the analysis of the information plane dynamics for each layer, which represents the dynamics of information between $Z$ and $X$, as well as between $Z$ and $Y$, layer-wise training exhibited uniform compression or improvement of both nHSIC(Z;X) and nHSIC(Z;Y) at each layer.
> In contrast, E2E training allowed different nHSIC behaviors at each layer, and in relatively deep networks, IB behavior was observed at the final layer. From this observation, this paper claims that the advantage of being E2E, i.e., enabling interaction among layers, is having different information-theoretical roles among layers and acquiring IB representation at the last layer.
>
> As pointed out, it was unclear why layer-wise training is suitable for the comparison of E2E training and what was concluded about the information bottleneck from experimental results. To clarify this point, we have updated the following points in the main text.
> - (In abstract, update) "a comparison with layer-wise training, **which shares fundamental learning principles and architectures with E2E training, with the granularity of loss evaluation being the only difference.**"
> - (In section 4.3, add to end) **"This fact implies that the advantages of being E2E, i.e., enabling interaction among layers, is having different information-theoretical roles among layers and acquiring information bottleneck representation at the last layer."**
>
> > 2.2. How does the local optimization phenomenon observed in layer-wise training support the IB theory of E2E training?
>
> We state that acquiring IB representation at the final layer requires interaction among layers, which is present in E2E training but absent in layer-wise training.
> This follows from the fact that the local optimization in layer-wise training shows a uniform information compression, i.e., a decrease in both nHSIC(Z; X) and nHSIC(Z; Y), at each layer, while E2E training achieves IB behavior, i.e. compressing only nHSIC(Z; X), at the final layer despite information compression in the middle layers.
>
> As for the issues of local optimization in layer-wise learning, two points are highlighted in our work. \
>   i)  it hinders the propagation of input information (see section 4.2).  \
>   ii) it leads to uniform behaviors of nHSIC(Z;X) and nHSIC(Z;Y) at each layer, failing to achieve various information-theoretic behaviors as in E2E training (see section 4.3 and 4.4). \
> Particularly regarding the second point, the uniform behavior of each layer brought by local optimization is disadvantageous for acquiring the IB representation at the final layer, as seen in E2E training where the model output behaves like IB while exhibiting different behaviors at middle layers.
> Intuitively, it can be interpreted that for identifying and reducing unnecessary information of input $X$ for solving task $Y$ that is contained in the final layer representation, the cooperative relationships among layers brought by backpropagation are necessary.
>
>
>  >  (Weakness 3) Instead of showing the HSIC of different epochs, it will be more straightforward to visualize those of different layers in the same plot. In the paper the x and y scales of different layers are not shared.
>
> There are two reasons why we show nHSIC values of different epochs at the fixed layer rather than nHSIC values for different layers at the fixed epoch. \
>    i) We focus on the training dynamics of nHSIC values, which is motivated by information plane analysis of information bottleneck work [3]. \
>    ii) The nHSIC estimator is a consistent estimator, but the estimates calculated from the mini-batch are basically a biased estimator. That is why the comparison of nHSIC estimates among different layers could be affected by the different expectations of nHSIC estimates and lead to wrong conclusions. Therefore, we avoided comparing measured values of nHSIC at the fixed epoch among layers, instead, we compared the training dynamics of nHSIC values among layers. Please note that the training dynamics of nHSIC values itself is the comparison of nHSIC values in the same layer.
>
> [3] Geiger, "On information plane analyses of neural network classifiers–a review", 2021.

---

### Author Response · Authors · 2024-05-02
**Paper Revision**

We have uploaded the revised version of the paper given the reviews.
We hope that all the required changes from reviewers have been incorporated.

---

### Decision · Action_Editor_QJQR · 2024-05-24

**Recommendation:** Accept as is

**Comment:**

Despite some concerns from the reviewers, they are unanimous in deciding that the paper meets both the claims and evidence and audience criteria.

For my part, I feel as though the paper does a good job of not overselling what it is.  The paper is a primarily experimental investigation and some associated commentary about those experiments.  TMLR is meant to be a home for such papers.

**Audience:**

The paper has a clear audience in TMLR.  People interested in layer wise training and some of the differences between that and end-to-end training will have things to think about.

**Claims And Evidence:**

There is consensus among the reviewers that the paper does provide accurate, convincing and clear evidence for its claims.

To its credit, the paper doesn't appear to make very controversial or bold claims, instead the primary claims made are that 1. layer-wise training causes degeneration in mutual information which they demonstrate convincingly in a toy model, and 2. that E2E training compresses middle representations while maintaining high HSIC with the final layer.  This is shown in several experiments.

I think the paper constitutes some interesting observations and a reasonable discussion of those observations.